# Adaptive Frontier Exploration on Graphs with Applications to Network-Based Disease Testing

**Davin Choo**[*]
Harvard University
davinchoo@seas.harvard.edu

**Yuqi Pan**[*]
Harvard University
yuqipan@g.harvard.edu

**Tonghan Wang**
Harvard University
twang1@g.harvard.edu

**Milind Tambe**
Harvard University
tambe@seas.harvard.edu

**Alastair van Heerden**
University of Witwatersrand
Wits Health Consortium
alastair.vanheerden@wits.ac.za

**Cheryl Johnson**
World Health Organization
johnsonc@who.int

## Abstract

We study a sequential decision-making problem on a $n$-node graph $\mathcal{G}$ where each node has an unknown label from a finite set $\mathbf{\Omega}$, drawn from a joint distribution $\mathcal{P}$ that is Markov with respect to $\mathcal{G}$. At each step, selecting a node reveals its label and yields a label-dependent reward. The goal is to adaptively choose nodes to maximize expected accumulated discounted rewards. We impose a frontier exploration constraint, where actions are limited to neighbors of previously selected nodes, reflecting practical constraints in settings such as contact tracing and robotic exploration. We design a Gittins index-based policy that applies to general graphs and is provably optimal when $\mathcal{G}$ is a forest. Our implementation runs in $\mathcal{O}(n^2 \cdot |\mathbf{\Omega}|^2)$ time while using $\mathcal{O}(n \cdot |\mathbf{\Omega}|^2)$ oracle calls to $\mathcal{P}$ and $\mathcal{O}(n^2 \cdot |\mathbf{\Omega}|)$ space. Experiments on synthetic and real-world graphs show that our method consistently outperforms natural baselines, including in non-tree, budget-limited, and undiscounted settings. For example, in HIV testing simulations on real-world sexual interaction networks, our policy detects nearly all positive cases with only half the population tested, substantially outperforming other baselines.

## 1 Introduction

We study a sequential decision-making problem on a graph $\mathcal{G}$, where each node has an unknown discrete label from $\mathbf{\Omega}$. The labels follow a joint distribution $\mathcal{P}$, which we assume is specified by a Markov random field (MRF) defined over $\mathcal{G}$ [KF09]. When we act on a node, its label is revealed and we receive a label-dependent reward. Crucially, the entire process is *history-sensitive*: label realizations are stochastic and depend on previously observed labels, a setting that naturally arises in Bayesian adaptive planning [GK11]. In this paper, we study a setting where actions are subject to a *frontier exploration constraint*: the first node in each connected component is selected based on a pre-defined priority rule, and subsequent actions are restricted to neighbors of previously selected nodes. This constraint reflects realistic settings where local neighborhood information becomes accessible only through exploration, as in active search on graphs [GKX+12], robotic exploration

---

[*]Equal contribution

39th Conference on Neural Information Processing Systems (NeurIPS 2025).

[KK14], and cybersecurity applications [LCH$^+$25]. The objective is then to maximize the expected accumulated discounted reward over time by sequentially selecting nodes to act upon.

**Definition 1** (The *Adaptive Frontier Exploration on Graphs* (AFEG) problem). An AFEG instance is defined by a triple $(\mathcal{G}, \mathcal{P}, \beta)$, where $\mathcal{G} = (\mathbf{X}, \mathbf{E})$ is a graph, $\mathcal{P}$ is a joint distribution over node labels that is Markov with respect to $\mathcal{G}$, and $\beta \in (0,1)$ is a discount factor. The process unfolds over $n = |\mathbf{X}|$ time steps, with the state $\mathcal{S}_t$ at time $t$ consisting of the current frontier and the revealed labels. Acting on a frontier node reveals its label, grants a label-dependent reward, and updates beliefs about other nodes via Bayesian inference under $\mathcal{P}$. The goal is to compute a policy $\pi$ that maps each state to a frontier node, maximizing the expected total discounted reward:

$$\pi^* = \arg\max_{\pi} \sum_{t=1}^{n} \beta^{t-1} \sum_{v \in \mathbf{\Omega}} \mathcal{P}(X_{\pi(\mathcal{S}_{t-1})} = v \mid \mathcal{S}_{t-1}) \cdot r(X_{\pi(\mathcal{S}_{t-1})}, v),$$

where $X_{\pi(\mathcal{S}_{t-1})}$ is the node selected by policy $\pi$ at time $t$, and $r(\cdot, \cdot)$ is the label-dependent reward.

While the optimal policy can be computed via dynamic programming, it is intractable for general graphs due to the exponential state space. A natural strategy is to leverage adaptive submodularity, which guarantees that greedy policies achieve a $(1 - 1/e)$-approximation [GK11]. Unfortunately, the objective in AFEG is not adaptively submodular in general: for instance, in disease detection, observing an infected neighbor can *increase* the marginal benefit of testing a node, violating the diminishing returns property of adaptive submodularity.

Our problem is closely related to the setting of active search on graphs [GKX$^+$12, WGS13, JMC$^+$17, JMA$^+$18], where the goal is to identify as many target-labeled nodes as possible under a fixed budget, without exploration constraints. Since exact optimization is intractable, these works focused on practical heuristics such as search space pruning. AFEG differs in two key respects: (i) we impose a frontier constraint, and (ii) we consider an infinite-horizon objective with discounting, rather than a fixed budget. These differences are not merely technical but they enable provable optimality in meaningful special cases, particularly when the input graph $\mathcal{G}$ is a forest. Forest structures naturally arise in several relevant domains, including transmission trees in contact tracing [KFH06] and recruitment trees in respondent-driven sampling [Hec97, GS09]. Moreover, algorithms with guarantees on forests can be efficiently applied to sparse real-world interaction graphs, such as sexual contact graphs, which tend to be tree-like in practice; see Section 4.3.

## 1.1 Motivating application: network-based disease testing

A key motivating example of AFEG is network-based infectious disease testing where the goal is to identify infected individuals as early as possible. In particular, we focus on diseases that are transmitted through person-to-person contact[2], e.g., sex, exposure of blood through injecting drug use, or birth, where interaction information can be collected through interviews. In this context, frontier testing is both natural and operationally motivated: test outcomes substantially alter beliefs about neighboring individuals, making sequential expansion along the frontier an efficient strategy.

**Public health motivation.** The 95-95-95 HIV[3] targets proposed by UNAIDS [UNA22] aim for 95% of people with HIV to know their status, 95% of those to receive treatment, and 95% of treated individuals to achieve viral suppression — aligned with UN Sustainable Development Goal 3.3 [Nat]. Yet, the 2024 UNAIDS report [UNA24] reveals that the "first 95" remains the most elusive, with roughly one in seven people living with HIV still undiagnosed, and there continues to be 1.3 million new infections every year. Studies have shown that virally suppressed individuals will not infect others [CCM$^+$11, RCB$^+$16, BPP$^+$18], leading to the U=U (undetectable = untransmittable) campaign [oAD19, OG20]. Thus, the faster we can detect infected individuals, the faster they can be enrolled onto treatment and limit the spread of the disease. To address this gap, the WHO recommends network-based testing strategies to reach underserved populations [Org24a]. These include partners and biological children of people with HIV, as well as those with high ongoing HIV risk. Network-based interventions have shown effectiveness in South Africa [JPC$^+$19] and have also been explored for other infectious diseases beyond HIV [JSK$^+$17, MWBDM$^+$25]; see also [CLJ$^+$24] for a WHO-commissioned systemic review on social network-based HIV testing.

---

[2]This is in contrast to illnesses like flu where transmission can occur to a room full of strangers.

[3]The human immunodeficiency virus (HIV) attacks the immune system and can lead to AIDS. It remains a major global health issue, having claimed over 42 million lives to date [Org24b].

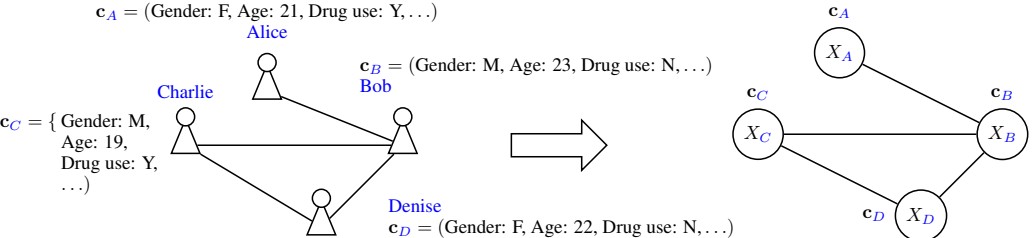

Figure 1: Illustration of how a real-world transmission graph (left) can be framed as an AFEG instance. Here, the joint distribution $\mathcal{P}$ over the labels $X_A, X_B, X_C, X_D \in \{+, -\}$ may depend on the covariates $\mathbf{c}_A, \mathbf{c}_B, \mathbf{c}_C, \mathbf{c}_D \in \mathbb{R}^d$ and underlying interaction graph structure.

Fig. 1 illustrates how we can model the network-based disease testing problem into a AFEG instance. Firstly, we use the network $\mathcal{G}$ as is, where nodes represent individuals and edges represent sexual interactions. Each node has a binary infection status (infected or not) that is drawn from some underlying joint distribution $\mathcal{P}$ on $\mathbf{X}$ over the labels $\mathbf{\Omega} = \{+, -\}$, where $\mathcal{P}$ may depend on the individual covariates and graph structure. The reward for testing individual $X$ and revealing status $b \in \{0, 1\}$ is then $r(X, b) = b$. See Fig. 1 for an illustration. The goal is of trying to identify infected individuals as early as possible is implicitly enforced by the presence of *any* discount factor $\beta < 1$. Importantly, discounting reflects both practical constraints – such as sudden funding cuts [UNA25] – and clinical importance of early diagnosis, which improves patient outcomes and limits transmission [CCM+11]. See also [RN21] for other natural justifications for using discount factors $\beta$ in modeling long-term policy rewards. While transmission graphs of sexually transmitted diseases are not truly forests and may have high-degree nodes (e.g., sex workers), empirical studies have also shown that such transmission graphs are often sparse and exhibit tree-like structure [BMS04, YJM+13, WKPF+17]. Finally, to apply the infinite horizon framework of AFEG in our finite testing setting, we give zero subsequent rewards after every individual has been already tested.

## 1.2 Our contributions

**Contribution 1: Gittins index-based policy for AFEG and new results for branching bandits.** In Section 3, we show that when $\mathcal{G}$ is a forest, AFEG can be modeled as a *branching bandit* problem, for which Gittins index policies are known to be optimal [KO03]. We provide a novel characterization of Gittins indices for discrete branching bandits using piecewise linear functions, and develop a practical implementation that runs in $\mathcal{O}(n^2 \cdot |\mathbf{\Omega}|^2)$ time while using $O(n \cdot |\mathbf{\Omega}|^2)$ oracle calls to $\mathcal{P}$ and $O(n^2 \cdot |\mathbf{\Omega}|)$ space. Our policy also works for general non-tree AFEG instances, but without optimality guarantees. Despite this, it demonstrates strong performance in experimental evaluations.

**Contribution 2: Formalizing network-based disease testing as an AFEG instance.** As shown in Section 1.1, network-based infectious disease testing can be cast as an instance of AFEG. To our knowledge, this is the first formal framework to model frontier-based testing as sequential decision-making on a probabilistic graph model for principled exploitation of network effects in diseases such as HIV. In Appendix C, we propose a method to learn parameters from past disease data to define a joint distribution $\mathcal{P}$ on new interaction networks so as to define new AFEG instances.

**Contribution 3: Empirical evaluation.** We evaluate our Gittins index-based policy on synthetic datasets and show that it performs strongly even in settings where it is not provably optimal, including non-trees and finite-horizon scenarios. Our approach outperforms other baselines on public-use real-world sex interaction graphs on 5 sexually transmitted diseases (Gonorrhea, Chlamydia, Syphilis, HIV, and Hepatitis) from ICPSR [MR11]. For instance, in one of our experiments on HIV testing (see Fig. 5), our method identifies almost all infected individuals while other baselines would only detect about 80%, in expectation, if we only have the testing budget to only test half of the population.

## 2 Preliminaries

**Notation.** We use lowercase letters for scalars, uppercase letters for random variables, bold letters for vectors or collections, and calligraphic letters for structured objects such as graphs and probability

distributions. Unordered sets are denoted with braces (e.g., $\{\cdot\}$), and ordered tuples with parentheses (e.g., $(\cdot)$). For any set $\mathbf{A}$, let $|\mathbf{A}|$ denote its cardinality. We use $\mathbb{R}_{\geq 0}$ for non-negative reals, $\mathbb{N}$ for the natural numbers, and $\mathbb{N}_{>0} = \mathbb{N} \setminus \{0\}$. For any $n \in \mathbb{N}_{>0}$, we define $[n] := \{1, \dots, n\}$. For a vector $\mathbf{x} = (x_1, \dots, x_n)$, we use $\mathbf{x}_i = x_i$ and $\mathbf{x}_{-i} = (x_1, \dots, x_{i-1}, x_{i+1}, \dots, x_n)$ to denote the vector without the $i$-th coordinate. We also employ standard asymptotic notations such as $\mathcal{O}(\cdot)$ and $\Omega(\cdot)$.

In this work, we consider joint distributions over $n$ discrete variables $\mathbf{X} = \{X_1, \dots, X_n\}$, structured by an undirected graph $\mathcal{G} = (\mathbf{X}, \mathbf{E})$. Each variable $X_i$ takes values from a finite set of labels $\boldsymbol{\Omega} = \{v_1, \dots, v_{|\boldsymbol{\Omega}|}\}$; in the binary case, $\boldsymbol{\Omega} = \{0, 1\}$. For any node $X \in \mathbf{X}$, let $\mathbf{N}(X) \subseteq \mathbf{X}$ denote its neighbors in $\mathcal{G}$, and let $\mathbf{V}(\mathcal{G})$ denote the vertex set. As standard in the literature of graphical models, we will use $X \in \mathbf{X}$ to refer to both the node and its associated random variable, so expressions like $X = v$ denote the event that variable $X$ takes on label value $v \in \boldsymbol{\Omega}$. A tree is a connected acyclic graph, and a forest is a collection of disjoint trees. A rooted tree designates one node as the root and orients all edges away from it. In a directed rooted tree, we denote the parent and children of $X$ by $\mathrm{Pa}(X)$ and $\mathrm{Ch}(X)$ respectively, with $\mathrm{pa}(X)$ as the realization of its parent(s). Note that in rooted trees, $\mathrm{Pa}(X) = \emptyset$ if and only if $X$ is the root. The most standard and general way to model a joint distribution $\mathcal{P}$ that is Markov with respect to a graph $\mathcal{G}$ is via a Markov Random Field (MRF) [KF09]. An MRF is an undirected graphical model in which nodes represent random variables and edges encode conditional dependencies. It satisfies the local Markov property: each variable is conditionally independent of all others given its neighbors. See Appendix A for additional background.

AFEG bears some resemblance to Bayesian multi-armed bandits (MABs), where Gittins index policies are optimal under assumptions like arm independence and infinite-horizon discounted rewards [Git79, GGW11]. However, key differences prevent a reduction of AFEG to a standard MAB: (i) each arm (node) can be selected at most once; (ii) the set of available actions changes dynamically due to the frontier constraint; and (iii) action outcomes are correlated through the graph structure and joint distribution $\mathcal{P}$. As such, a closer abstraction is the *branching bandit* model [Wei88, Tsi94, KO03], where actions dynamically activate new options, closely mirroring frontier expansion in AFEG. While Gittins index policies are known to be optimal for branching bandits [KO03], no efficient method has been proposed to compute them in general. Indeed, Gittins indices are underused in practice due to perceived computational intractability in all but simple settings [Sco10, MKLL12, Edw19]. Our work addresses this gap by presenting the first efficient implementation of Gittins-based policies in discrete branching bandits with history-dependent rewards, enabling their use in structured settings like network-based disease testing. Related index-based techniques for classic MABs are reviewed in [CM14], but do not extend to branching structures. [MK23] studied a disease-spread model in which infections propagate sequentially along parent-child links from a known root, and showed that it reduces to the branching-bandit model of [Wei88]. In contrast, our setting assumes no known source: infection statuses are jointly distributed over the network, and testing can start at any node under a frontier constraint. This makes their model ill-suited for our AFEG setting, where correlations cannot be captured by a single infection tree.

Our work is also related to well-studied areas such as reinforcement learning, active search on graphs, and influence maximization. In addition, there is a substantial body of prior work on network-based HIV testing and transmission modeling. We provide a detailed review of these topics in Appendix A.

## 3 A Gittins index-based policy for the AFEG problem

We propose a policy, GITTINS, for the AFEG problem that is based on Gittins indices. In Section 3.1, we show that when the input graph $\mathcal{G}$ is a forest, AFEG reduces exactly to the branching bandit framework [Wei88, Tsi94, KO03], under which GITTINS is provably optimal. While [KO03] established the existence of an optimal Gittins index policy, they did not characterize the index explicitly nor provide an efficient method for computing it. In Section 3.2, we prove that the key recursive functions involved in computing the index are piecewise linear, which enables practical and efficient computation. The implementation of GITTINS and its runtime analysis is given in Section 3.3.

### 3.1 Reduction to branching bandits for tree-structured instances

In the branching bandit model [Wei88, Tsi94, KO03], a project is represented as a rooted tree, where each node corresponds to an action. Selecting a node yields a stochastic immediate reward and activates its children to be available for future selection. A node is available if it is the root or a

descendant of a previously selected node. Under the frontier exploration constraint of AFEG, a forest $\mathcal{G}$ naturally induces a collection of rooted trees; see Fig. 2. The problem then reduces to a branching bandit instance by treating each leaf as if it can be selected infinitely many times with zero reward, or equivalently, by appending an infinite chain of zero-reward nodes to each leaf. At each timestep, the available actions correspond to the current frontier: nodes adjacent to those already selected. Crucially, due to the Markov property of $\mathcal{P}$, selecting a node $X$ only affects posterior beliefs over its descendants, preserving the conditional independence structure required by the branching bandit formulation. When $\mathcal{G}$ is a forest, the problem decomposes across connected components, allowing Gittins indices to be computed independently for each rooted tree.

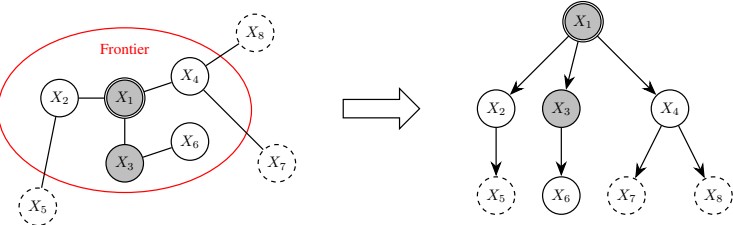

Figure 2: Reduction to a branching bandit on 8 nodes with root $X_1$. After acting on $\{X_1, X_3\}$, the frontier is $\{X_2, X_4, X_6\}$. Note that we have $\mathcal{P}(x_2 \mid x_1, x_3) = \mathcal{P}(x_2 \mid x_1)$ by the Markov property.

To define the Gittins index, we assume the reward $r(X, v)$ for revealing label $v \in \mathbf{\Omega}$ at node $X$ is bounded by some finite $\bar{r}$, i.e., $-\infty < -\bar{r} \leq r(X, v) \leq \bar{r} < \infty$, and introduce a retirement option with one-off reward $m$.[4] This enables a recursive characterization of the index: at any step, the policy can choose to continue exploring, or stop acting and receive an one-off fixed reward of $m$. Since rewards are upper bounded by $\bar{r}$, the maximum attainable reward is at most $\bar{r} + \beta\bar{r} + \beta^2\bar{r} + \ldots = \frac{\bar{r}}{1-\beta}$. So, when $m > \frac{\bar{r}}{1-\beta}$, any optimal policy should choose the retirement action and quit.

To define the Gittins index, let us first define two recursive functions $\phi$ and $\Phi$, as per [KO03]. For any non-root node $X \in \mathbf{X}$, label $b \in \mathbf{\Omega}$, and value $0 \leq m \leq \frac{\bar{r}}{1-\beta}$,

$$\phi_{X,b}(m) = \max\left\{m, \sum_{v \in \mathbf{\Omega}} \mathcal{P}(X = v \mid \mathrm{Pa}(X) = b) \cdot \left[r(X, v) + \beta \cdot \Phi_{\mathrm{Ch}(X),v}(m)\right]\right\} \quad (1)$$

If $X$ is the root, we define $\phi_{X,\emptyset}(m) = \max\left\{m, \sum_{v \in \mathbf{\Omega}} \mathcal{P}(X = v) \cdot \left[r(X, v) + \beta \cdot \Phi_{\mathrm{Ch}(X),v}(m)\right]\right\}$.

For any subset of nodes $\mathbf{S} \in \mathbf{X}$, label $v \in \mathbf{\Omega}$, and value $0 \leq m \leq \frac{\bar{r}}{1-\beta}$,

$$\Phi_{\mathbf{S},v}(m) = \begin{cases} \frac{\bar{r}}{1-\beta} - \int_m^{\frac{\bar{r}}{1-\beta}} \prod_{Y \in \mathbf{S}} \frac{\partial \phi_{Y,v}(k)}{\partial k} \, dk & \text{if } \mathbf{S} \neq \emptyset \\ m & \text{if } \mathbf{S} = \emptyset \end{cases} \quad (2)$$

We will only invoke Eq. (2) with $\mathbf{S} = \mathrm{Ch}(X)$ for some node $X$. The interpretation here is that $\phi_{X,b}$ represents the total expected value of this subtree rooted at $X$ when its parent $\mathrm{Pa}(X)$ has label $b \in \mathbf{\Omega}$, while accounting for option of taking the retirement option $m$ at each step, and $\Phi_{\mathrm{Ch}(X),v}$ represents the value of the collection of subtrees rooted at the children of $X$, *excluding $X$ itself*, i.e., the contributions from the children of $X$, conditioned on $X$ having label $b \in \mathbf{\Omega}$. For example, for $X_1$ in Fig. 2, this refers to the subtrees rooted at $X_2$, $X_3$, and $X_4$. Using these notation, the Gittins index $g(X, b)$ for node $X$ given $\mathrm{Pa}(X) = b$ is then defined as

$$g(X, b) = \min\left\{m \in \left[0, \frac{\bar{r}}{1-\beta}\right] : \phi_{X,b}(m) \geq m\right\} \quad (3)$$

That is, $g(X, b)$ represents the "fair value" of the subtree rooted at $X$, given that its parent has label $b \in \mathbf{\Omega}$. The parent's label matters because the posterior distribution over $X$'s label is updated conditionally based on the value of $b$. Theorem 3, which follows from Theorem 1 of [KO03] with appropriate changes in notation, establishes that the GITTINS policy is optimal when $\mathcal{G}$ is a forest.

---

[4]There are several equivalent ways of proving the optimality of Gittins indices in the classic non-branching setting, e.g., the original stopping problem formulation [Git74], retirement option process formulation [Whi80], restart-in-state formulation [KVJ87], prevailing charge formulation [Web92], state space reduction [Tsi94], etc. The branching bandit optimality proof of [KO03] builds on [Whi80]'s retirement option formulation.

**Definition 2** (The GITTINS policy; Algorithm 1). *The* GITTINS *policy pre-computes all* $g(X, b)$ *values given* $\mathcal{G}$ *and* $\mathcal{P}$*, then repeatedly acts on the node in the frontier with the largest index value.*

**Theorem 3.** GITTINS *is optimal for the* AFEG *problem when* $\mathcal{G}$ *is a forest.*

### 3.2 Properties of discrete branching bandits

We prove that the recursive functions $\phi_{X,b}(m)$ and $\Phi_{\mathrm{Ch}(X),v}(m)$ are piecewise linear in $m$. This facilitates an efficient implementation of GITTINS, which we give in Algorithm 1.

**Lemma 4.** *For any node* $X \in \mathbf{X}$ *and label* $b \in \mathbf{\Omega}$*,* $\phi_{X,b}(m)$ *is a non-decreasing piecewise linear function over* $m \in [0, \bar{r}/(1 - \beta)]$*.*

The proof intuition behind Lemma 4 is to perform induction from the leaves to the root while recalling that piecewise functions on a fixed domain range are closed under addition, multiplication, differentiation, and integration. This later allows us to bound the running time for computing our GITTINS policy in Theorem 6 as the number of pieces in the function changes additively as we combine piecewise functions, e.g. $\#\mathrm{pieces}(f_1 + f_2) \le \#\mathrm{pieces}(f_1) + \#\mathrm{pieces}(f_2)$.

We also prove additional properties of the $\Phi$ function which are crucial in our algorithm for efficiently manipulating the piecewise linear functions to compute the recursive functions $\phi$ and $\Phi$, and may be of independent interest to resaerchers of Gittins index.

**Proposition 5.** *For any non-leaf node* $X$ *and label* $b$*:*

- $\Phi_{\mathrm{Ch}(X),v}(m) = \Phi_{\mathrm{Ch}(X),v}(0) + h_{\mathrm{Ch}(X),v}(m)$ *for some piecewise linear* $h_{\mathrm{Ch}(X),v}(m)$*.*

- $\Phi_{\mathrm{Ch}(X),v}(m) = m$ *if and only if* $m \ge \max_{Y \in \mathrm{Ch}(X)} g(Y, b)$*.*

The first term in the expression $\Phi_{\mathrm{Ch}(X),v}(m) = \Phi_{\mathrm{Ch}(X),v}(0) + h_{\mathrm{Ch}(X),v}(m)$ can be interpreted as the original maximized reward for the descendants of $X$ while the second term is the additional reward afforded by the retirement option $m$. Meanwhile, we observe that the minimum $m$ such that $\Phi_{\mathrm{Ch}(X),v}(m) = m$ is the essentially the largest Gittins index among nodes in $\mathrm{Ch}(X)$.

Full proofs of Lemma 4 and Proposition 5 are deferred to Appendix F.1.

### 3.3 Extension to general graphs

On general graphs where $\mathcal{G}$ is not a forest, dependencies across frontier nodes may violate the branching bandit assumptions. Nevertheless, we heuristically apply GITTINS by treating each connected component as a tree (e.g., a minimum spanning tree) and ignoring edges that violate the acyclicity requirement. This restricts the frontier to a subtree at each step.

Algorithm 1 provides a pseudocode describing this, where Line 5 is the heuristic which drops edges when $\mathcal{G}$ is not a forest graph. As any tree restriction works here, one natural option is to compute the breadth-first search (BFS) tree as it minimizes the height to the root, reducing any artificial frontier constraint due to tree projection. The runtime complexity of Algorithm 1 is given in Theorem 6.

At first glance, one may think that the running time of Algorithm 1 would dependent exponentially on the maximum depth $d$ of the induced rooted trees, even if all operations involving piecewise linear functions can be done in $O(1)$ time. This is because the definitions of Eq. (1) and Eq. (2) tell us that the function $\phi_{X,b}$ depends on all $\phi_{Z,v}$ functions, for all descendants $Z$ of $X$ and labels $v \in \mathbf{\Omega}$. However, our next result show that we can in fact obtain a polynomial run time that is *independent* of the maximum depth $d$ of the induced rooted trees; this is why *any* tree restriction works on Line 5.

**Theorem 6.** *Given graph* $\mathcal{G} = (\mathbf{X}, \mathbf{E})$ *and oracle access[5] to joint distribution* $\mathcal{P}$*, the Gittins indices can be computed in* $O(n^2 \cdot |\mathbf{\Omega}|^2)$ *time while using* $O(n \cdot |\mathbf{\Omega}|^2)$ *oracle calls to* $\mathcal{P}$ *via* Algorithm 1*. The space complexity is* $O(n^2 \cdot |\mathbf{\Omega}|)$ *space for storing* $O(n \cdot |\mathbf{\Omega}|)$ *intermediate piecewise linear functions.*

The proof outline of Theorem 6 is as follows: we first use induction (from the leaves towards the root) to argue that, for any node $X$, the set of functions $\{\phi_{X,b}\}_{b \in \mathbf{\Omega}}$ can be computed using $O(|\mathbf{\Omega}|^2)$

---

[5]The focus is on the recursive cost of computing Gittins indices and the oracle access assumption is meant to abstract away the computational cost of computing quantities like $\mathcal{P}(\cdot \mid \cdot)$. As our setting assumes that $\mathcal{P}$ is a pairwise MRF defined over a tree-structured graph, such inference is tractable via the junction tree algorithm.

---
**Algorithm 1** Setting up the GITTINS policy.

---
1: **Input**: Graph $\mathcal{G} = (\mathbf{X}, \mathbf{E})$, Joint distribution $\mathcal{P}$ over $\mathbf{X}$ that is Markov to $\mathcal{G}$
2: **Output**: Gittins index value $g(X, b)$ for all $X \in \mathbf{X}$ and $b \in \Omega$
3: **for** each connected component $\mathcal{H}$ of $\mathcal{G}$ **do**
4:     Compute root node $X_{root}$ from priority rule    $\triangleright$ e.g., $X_{root} = \operatorname{argmax}_{X_i \in \mathbf{V}(\mathcal{H})} \mathcal{P}(X_i = 1)$
5:     Compute *any* tree of $\mathcal{H}$ rooted at $X_{root}$             $\triangleright$ Heuristic for non-trees
6:     **for** node $X_i \in \mathbf{V}(\mathcal{H})$ from leaf towards root, and label $b \in \Omega$ **do**    $\triangleright$ For $X_{root}$, set $b = \emptyset$
7:         **if** $X_i$ is a leaf **then**
8:             Compute $\phi(X_i, b)(m) = \max\left\{m, \beta m + \sum_{v \in \Omega} \mathcal{P}(X = v \mid \operatorname{Pa}(X) = b) \cdot r(X, v)\right\}$
9:         **else**
10:             Compute $\Phi_{\operatorname{Ch}(X_i),v}(m)$ via Eq. (2) and Proposition 5
11:             Compute $\phi_{X_i,b}(m)$ via Eq. (1)
12:         Store $\phi(X_i, b)$ for future computation
13: Compute $\{g(X, b)\}_{X \in \mathbf{X}, b \in \Omega}$ according to Eq. (3), using previously stored $\phi$ values
14: **return** $\{g(X, b)\}_{X \in \mathbf{X}, b \in \Omega}$         $\triangleright$ Reminder: For root nodes, we set $b = \emptyset$

---

oracle calls to $\mathcal{P}$ and $O(|\Omega| \cdot \max\{1, |\operatorname{Ch}(X)|\})$ operations on piecewise linear functions, as long as we store intermediate functions $\{\phi_{Y,v}\}_{Y \in \operatorname{Ch}(X), v \in \Omega}$ along the way. Then, we argue that the maximum time to perform any piecewise linear function operation in Algorithm 1 is upper bounded by $O(n \cdot |\Omega|)$. Our claim follows by summing over all nodes $X$ and using the upper bound cost of operating on piecewise linear function. See Appendix F.1 for the full proof.

We also provide a worked example of how to compute Gittins index by manipulating $\phi$ and $\Phi$ in Appendix B. This computation is implemented Python and the source code is available on our Github.

## 4 Experiments

We benchmark our proposed GITTINS policy against several natural baselines — RANDOM, GREEDY, DQN, and OPTIMAL — on both synthetic and real-world graphs to evaluate performance on AFEG. To reflect the network-based disease testing application discussed in Section 1.1, we consider binary node labels, and define the immediate reward to be 1 if and only if the revealed label is positive. As such, it is natural to define the first node in every connected component as the node with the highest marginal probability of being positive amongst all nodes in that connected component.

**Benchmarked policies.** Given a problem instance $(\mathcal{G}, \mathcal{P}, \beta)$, a state in AFEG consists of the current set of frontier nodes and the revealed labels of previously tested nodes.
- RANDOM: Selects a random node from the frontier without using any state information.
- GREEDY: Selects the frontier node with the highest posterior probability of being positive, conditioned on the currently observed labels.
- DQN: Implements a deep Q-network baseline [MKS$^+$15], using the NNConv architecture from PyTorch Geometric [FL19]. This model applies a message-passing GNN with edge-conditioned weights [GSR$^+$17] to capture graph structure and node covariates.
- OPTIMAL: Computes the action that maximizes the expected total discounted reward for each possible state via brute-force dynamic programming. This method is tractable only on small graphs due to the combinatorial explosion of the state space.
- GITTINS: Our proposed method, described in Algorithm 1, which is provably optimal when the underlying graph $\mathcal{G}$ is a forest. We use breadth-first search (BFS) trees in Line 5.

Since AFEG and these policies are agnostic to how the joint distribution $\mathcal{P}$ is defined, we defer the details of how $\mathcal{P}$ is defined to Appendix E. That is, one may read the experimental section assuming access to some $\mathcal{P}$ oracle. For reproducibility, all code is provided on our Github repository[6].

### 4.1 Experiment 1: On tree inputs, GITTINS works well even in finite horizon settings

We evaluated the policies against a family of randomly generated synthetic trees on $n \in \{10, 50, 100\}$ nodes across various discount factors $\beta \in \{0.5, 0.7, 0.9\}$. We only run OPTIMAL for small $n = 10$

---

[6]https://github.com/cxjdavin/adaptive-frontier-exploration-on-graphs

instances, where the plots for OPTIMAL and GITTINS exactly overlap as expected. For $n = 10$, we exactly compute the expectation by weighting accumulated discounted rewards of each of the $2^{10}$ realizations by its probability. For $n \in \{50, 100\}$, we compute Monte Carlo estimates by sampling 200 random realizations from $\mathcal{P}$. For each setting of $(n, \beta)$, we generated 10 random trees and plot the mean ($\pm$ std. err.) of the expected accumulated discounted rewards over time for each policy. Fig. 3 shows a subset of our results for $\beta = 0.9$; see Appendix E for the full experimental results.

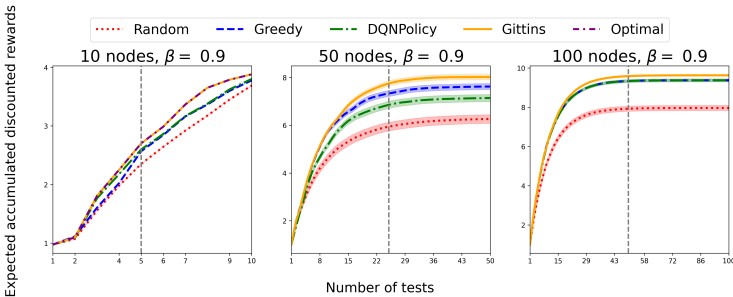

Figure 3: Subset of synthetic tree input results. GITTINS consistently beats other baselines at every fixed budget, e.g., vertical line indicates performance when only half the nodes can be acted upon.

Interestingly, while GITTINS is only proven to be optimal with respect to the expected accumulated discounted rewards, i.e., the rightmost slice of each plot, we see that it consistently outperforms all other baselines at every fixed timestep. For example, if we only have a fixed budget of being only to act on half the nodes (visualized by drawing a vertical line at the midpoint of the experiment), the GITTINS plot lies above the others.

## 4.2 Experiment 2: Gittins performance degrades gracefully for non-trees

Here, we investigate the degradation of GITTINS as a heuristic as the input graph deviates from a tree in a controlled manner. Using the same setup as Section 4.1, we see that the attained expected accumulated discounted reward of GITTINS degrades relative to GREEDY as we add more edges to a tree, as expected. Note that in the rightmost graph of Fig. 4, 10 additional edges is more than 20% additional edges compared to the original $n - 1 = 49$ tree edges.

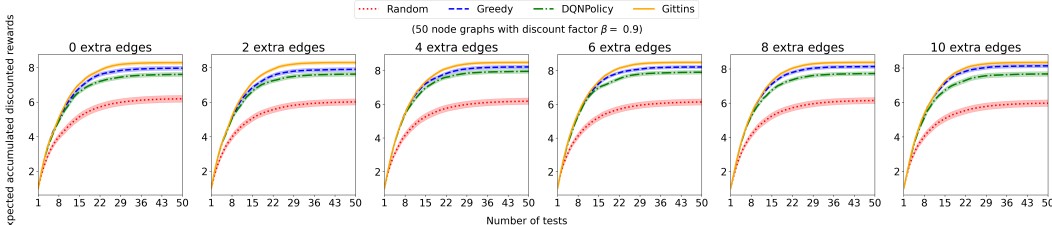

Figure 4: Synthetic experiment: the initial performance gains of GITTINS over GREEDY and DQN diminishes as we progressively add edges to 10 random 50-node trees with discount factor $\beta = 0.9$.

## 4.3 Experiment 3: Real-world sexual interaction graphs

Beyond experiments on synthetic graphs, we also evaluated the policies on real-world sexual interaction networks derived from a de-identified, public-use dataset released by ICPSR [MR11][7]. This dataset was originally collected to examine how partnership networks influence the transmission of sexually transmitted and blood-borne infections. It includes reported sexual edges, covariates of each individuals (e.g., whether the individual is unemployed or homeless, etc), and reported statuses for 5 sexually transmitted diseases: Gonorrhea, Chlamydia, Syphilis, HIV, and Hepatitis.

---

[7]This de-identified, public-use dataset is publicly available for download at `https://www.icpsr.umich.edu/web/ICPSR/studies/22140` upon agreeing to ICPSR's terms of use. IRB is not required; see Appendix E.

For each disease, we fit parameters as described in Appendix C and use these parameters to define $\mathcal{P}$ for each AFEG instance on these real-world sexual interaction graphs. To be precise, we define $\mathcal{P}$ to be a pairwise MRF over the transmission network $\mathcal{G}$, and $\mathcal{P}$ is parameterized by vectors $\theta_1 \in \mathbb{R}^{2+2d}$ and $\theta_2 \in \mathbb{R}^{4+5d}$, where $d \in \mathbb{N}$ is the the dimension of the covariates. Since the role of the discount factor $\beta \in (0,1)$ in AFEG is to encourage early identification of infected individuals, any value in that range is technically valid. To better reflect practical scenarios where timely detection is important across the entire population, we use $\beta = 0.99$ in our experiments and report results in terms of fraction of positive cases detected. In Appendix E, we explain in further detail how we pre-process the dataset, produce joint distributions $\mathcal{P}$ for each graph for the experiments.

Our experimental results on real-world graphs is given in Fig. 5.[8] As these graphs are large, we do not run OPTIMAL and use 200 Monte Carlo simulations to estimate the expected performance of each policy. To enable evaluation on larger graphs while preserving network structure, we applied a principled subsampling strategy based on connected components to preserve topological properties of the original network more faithfully than node-level subsampling. For each disease-specific dataset, we randomly shuffled the connected components and then greedily aggregated them into a new graph until the total number of nodes exceeded a specified threshold $\tau$. Due to the sparsity of the graphs, the resulting subsampled graphs contain approximately $\tau$ nodes as desired. To balance computational feasibility with dataset representativeness, we set $\tau = 300$. For the graph statistics of the full and subsampled real-world interaction graphs, see Table 1 and Table 2 in Appendix E.

Throughout all experiments, we consistently see that GITTINS outperforms or is competitive with the other baselines both in terms of expected fraction of positive cases detected and for any fixed timestep, even when the input is not a forest. For example, when limited to testing only half the population in the HIV experiment, GITTINS identifies nearly all infected individuals whereas other baselines detect only about 80%, in expectation. In terms of running time, observe that GREEDY and DQN become computationally intractable on large real-world graphs.[9] DQN incurs significant training overhead due to fitting a graph neural network for each $(\mathcal{G}, \mathcal{P}, \beta)$ instance while GREEDY is computationally expensive during rollout, requiring $\sum_{i=1}^{n} (n - i + 1) \in O(n^2)$ calls to the $\mathcal{P}$ oracle per Monte Carlo sample.[10] In contrast, GITTINS is efficient in both policy training (index computation) and rollout (selecting the frontier node with highest index), making it highly practical for real-world instances.

### 4.3.1 Noisy access to the underlying distribution $\mathcal{P}$

In practice, we do not have access to the true underlying distribution $\mathcal{P}$. Instead, we would likely only obtain an noisy version $\mathcal{Q}$ of $\mathcal{P}$ and have to make adaptive testing decisions based on $\mathcal{Q}$. Recalling $\mathcal{P}$ is parameterized by vectors $\theta_1$ and $\theta_2$, we define a noisy distribution $\mathcal{Q}_\varepsilon$ defined by noisy versions $\widetilde{\theta}_1$ and $\widetilde{\theta}_2$ of $\theta_1$ and $\theta_2$, where we add random noise that scales proportionally with the magnitude of each coordinate In the following experiments for $\varepsilon \in \{0, 0.25, 0.5, 0.75, 1\}$. That is, for a parameter value of $x$, we add a random noise of $[-\varepsilon x, \varepsilon x]$ to it. Fig. 6 illustrates the empirical performance of various policies on the HIV dataset when they only have access to $\mathcal{Q}_\varepsilon$ while the Monte Carlo evaluation and evolution of the statuses are based on $\mathcal{P}$; see Appendix E for the plots for all the other diseases. Unsurprisingly, all policies degrade towards the performance of the RANDOM baseline as $\varepsilon$ increases since $\mathcal{Q}_\varepsilon$ becomes less informative with respect to the true underlying distribution $\mathcal{P}$.

## 5 Conclusion and discussion

We introduced and studied the adaptive frontier exploration on graphs problem (Definition 1), a framework for sequential decision-making with label-dependent rewards under a frontier exploration constraint. Our Gittins index-based policy (Algorithm 1) is provably optimal on trees, runs in polynomial time, and demonstrates strong empirical performance on general graphs.

---

[8]All experiments were performed on a personal laptop (Apple MacBook 2024, M4 chip, 16GB memory).

[9]This is why we had to subsample up to $\tau = 300$ nodes when performing our empirical evaluation. RANDOM and GITTINS are able to run fast on larger graphs but comparing only these two policies is not interesting.

[10]Even if we optimize the implementation of GREEDY to only recompute marginal positive probabilities for nodes in the same connected component as the previously tested node, it still incurs a huge rollout time on graphs with large connected components. The timings in Fig. 5 are for this optimized GREEDY implementation.

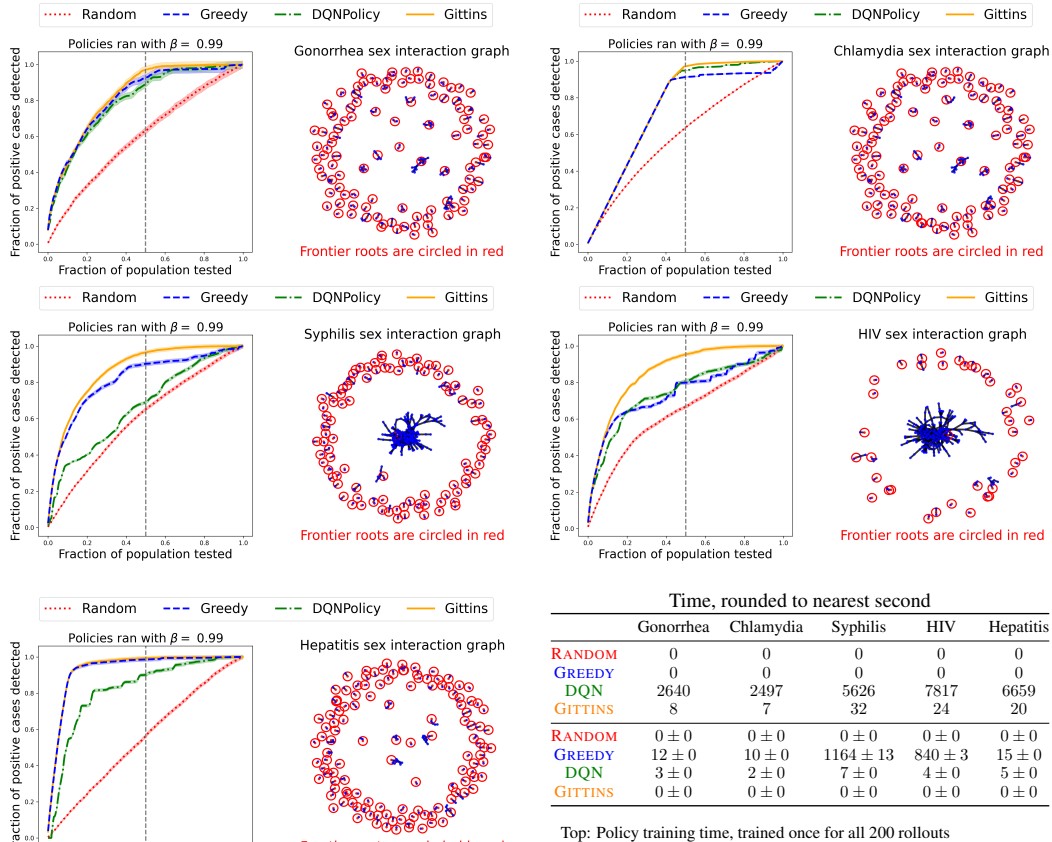

Figure 5: Experimental results for Gonorrhea, Chlamydia, Syphilis, HIV, and Hepatitis from sub-sampling connected components till we have at least 300 nodes. The vertical dashed line indicates performance when only half the individuals can be tested.

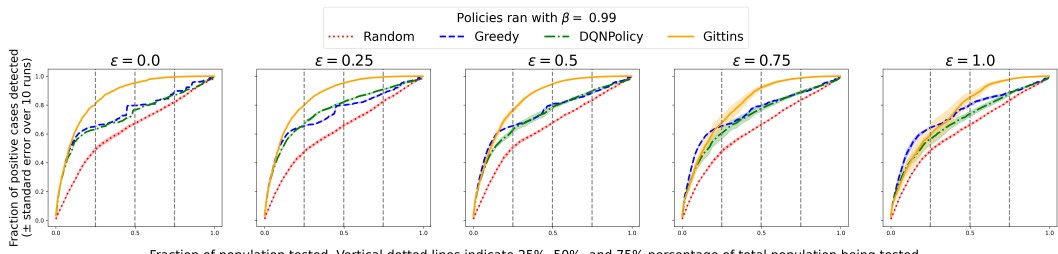

Figure 6: Experiments on the HIV dataset where policies only have access noisy version $\mathcal{Q}_\varepsilon$ of the underlying distribution $\mathcal{P}$ on the HIV dataset. Error bars illustrate the standard error due to 10 random instantiations of $\mathcal{Q}_\varepsilon$ for each corresponding value of $\varepsilon$.

**Broader impact and fairness.** This work is motivated by public health challenges, where limited resources and reduced funding [UNA25] highlight the need for more efficient testing strategies. The AFEG framework supports targeted, adaptive exploration of interaction networks, guided by a joint distribution $\mathcal{P}$ can incorporate domain knowledge. It also enables fairness-aware interventions through reward shaping, allowing practitioners to prioritize specific subpopulations within the same decision-making framework. Our proposed Gittins index-based policy operates within this flexible setup, making it suitable for responsible and context-aware deployment. Additional discussion of limitations and fairness considerations is provided in Appendix D.

## Acknowledgments and Disclosure of Funding

This work was supported by ONR MURI N00014-24-1-2742. The findings and conclusions in this report are those of the authors and do not necessarily represent the official position of the WHO. Davin would like to thank Bryan Wilder, Amulya Yadav, and Chun Kai Ling for their thought-provoking technical discussions, and Eric Rice, Geoff Garnett, Samuel R. Friedman, and Ashley Buchanan for generously sharing their domain expertise on HIV testing and transmission.

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

# A  Further related work

**Markov Random Fields (MRF).** By the Hammersley-Clifford theorem [HC71, Cli90], an MRF has the form: $\mathcal{P}(\mathbf{x}) = \frac{1}{Z} \prod_{\mathbf{C} \in \mathcal{C}} \psi_{\mathbf{C}}(\mathbf{x}_{\mathbf{C}})$, where $\mathcal{C}$ is the set of cliques in $\mathcal{G}$, $\psi_{\mathbf{C}}$ is a non-negative potential function over clique $\mathbf{C}$, $\mathbf{x}_{\mathbf{C}}$ is the realization of nodes in $\mathbf{C}$, and $Z$ is the normalizing constant. Alternatively, MRFs can be represented using factor graphs, which are bipartite graphs connecting variable nodes to factor nodes. Each factor $f_j$ maps a subset $\mathbf{X}_j$ of variables to a non-negative value. The joint distribution is then $\mathcal{P}(\mathbf{x}) = \frac{1}{Z} \prod_{j=1}^{m} f_j(\mathbf{x}_j)$. While both representations are equivalent, factor graphs make inference structure explicit and are often used in algorithmic implementations. Exact inference in MRFs is known to be intractable in general, with complexity scaling exponentially in the treewidth of $\mathcal{G}$ [WJ08, KF09]. While the definition of AFEG is does *not* necessitate the use of MRFs in general, any joint distribution $\mathcal{P}$ that is Markov with respect to $\mathcal{G}$ works, we find that it is a reasonable model in our disease testing application (Section 1.1) where real-world sex interaction graphs are often of low treewidth (see Section 4).

**Reinforcement learning (RL).** Sequential decision-making is classically modeled using Markov decision processes (MDPs) [Put14], and solved using reinforcement learning (RL) techniques. Prominent algorithms include Q-learning [WD92], policy gradient methods [SMSM99], and deep RL approaches like deep Q-networks (DQN) [MKS+15]. In principle, AFEG can be cast as an MDP, but doing so leads to an exponentially large state space: the agent must track both which nodes have been selected and their revealed labels. This complexity makes direct application of off-the-shelf RL methods impractical in our setting with customization and heavy finetuning.

**Active search on graphs.** As discussed in Section 1, AFEG is related to active search on graphs [GKX+12, WGS13, JMC+17, JMA+18], which aims to identify as many target nodes as possible under a budget. However, these works do not impose a frontier constraint while searching over binary labels, and typically assume a relaxed Gaussian random field model [ZLG03] for tractable inference. In contrast, our formulation models the joint distribution explicitly as an MRF. While exact inference is generally intractable, many real-world graphs — especially sexual contact networks — have low treewidth, making structured modeling with MRFs feasible in practice (see Section 4).

**Influence maximization.** Another well-studied sequential decision problem on graphs is influence maximization, where the goal is to select seed nodes to maximize influence spread under stochastic propagation models such as the independent cascade or linear threshold [KKT03]. This framework has been applied to health interventions, such as selecting peer leaders to disseminate information in HIV prevention efforts among homeless youth [YWR+17, WOVH+18]. While both influence maximization and AFEG involve decisions on graphs, their objectives differ: the former focuses on maximizing long-term diffusion, whereas AFEG emphasizes label-driven, reward-maximizing sequential actions under uncertainty and exploration constraints.

**Network-based HIV testing and transmission modeling.** HIV transmission dynamics have been extensively studied through network-based models, where individuals are represented as nodes and edges denote reported sexual or social contacts [Rot09, MGDGO25]. Such networks can constructed from contact tracing, respondent-driven sampling (RDS), or molecular surveillance data [Hec97, GS09, ADGR+19]. While existing research have used methods such as generalized estimating equations, mixed effects regression, graph attention networks have been used to fit transmission probabilities using parameters [BMAK+21, WLK+23, XFL+21], they do not model and consider the sequentiality of assigning tests to individuals.

# B  Worked example of Gittins computation for branching bandits

Recall the recursive formulas of $\phi$ and $\Phi$ for the Gittins index computation, from Eq. (1) and Eq. (2) respectively. For convenience, we reproduce them below. For $0 \le m \le \frac{\bar{r}}{1-\beta}$,

$$\phi_{X,b}(m) = \max \left\{ m, \sum_{v \in \mathbf{\Omega}} \mathcal{P}\left(X = v \mid \mathrm{Pa}(X) = b\right) \cdot \left[r(X,v) + \beta \Phi_{\mathrm{Ch}(X),v}(m)\right] \right\}$$

$$\Phi_{\mathbf{S},v}(m) = \begin{cases} \frac{\bar{r}}{1-\beta} - \int_{m}^{\frac{\bar{r}}{1-\beta}} \prod_{Y \in \mathbf{S}} \frac{\partial \phi_{Y,v}(k)}{\partial k} \, dk & \text{if } \mathbf{S} \ne \emptyset \\ m & \text{if } \mathbf{S} = \emptyset \end{cases}$$

These $\phi_{X,b}(m)$ values, for all labels $b \in \mathbf{\Omega}$, can be pre-computed before we execute our Gittins method via recursion from the leaves. Then, the corresponding Gittins value for node $X$ given a realized parent value of $b \in \mathbf{\Omega}$ is simply $\min\{m \in [0, \frac{\bar{r}}{1-\beta}] : \phi_{X,b}(m) \geq m\}$.

In the rest of this section, we will work through the Gittins index computation on the rooted tree $\mathcal{G} = (\mathbf{X}, \mathbf{E})$ shown in Fig. 7 with $\beta = 0.9$ and the following joint distribution

$$\mathcal{P}(\mathbf{X} = \mathbf{x}) = \frac{1}{Z} \prod_{(X_i, X_j) \in \mathbf{E}} \exp\left(\mathbb{1}_{x_i = x_j}\right) \tag{4}$$

where $Z$ is the normalizing constant and $\mathbb{1}_{x_i = x_j}$ is the indicator function of whether the realized values of $X_i$ and $X_j$ agree. In terms of our pairwise MRF representation, $\mathcal{P}$ can be represented with $\theta_{\text{unary}} = (0, 0)$ and $\theta_{\text{pairwise}} = (0, 1, 0, 1)$. For a non-root node $X \in \mathbf{X}$, we have

$$\mathcal{P}(X = 1 \mid \text{Pa}(X) = 0) = \frac{1}{1 + e} \tag{5}$$

$$\mathcal{P}(X = 1 \mid \text{Pa}(X) = 1) = \frac{e}{1 + e} \tag{6}$$

With $\beta = 0.9$ and $\bar{r} = 1$, we have $0 \leq m \leq \frac{\bar{r}}{1-\beta} = 10$.

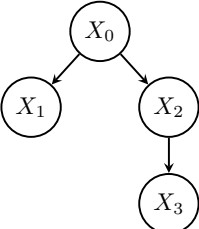

Figure 7: Example rooted tree $\mathcal{G} = (\mathbf{X}, \mathbf{E})$ over 4 nodes $\mathbf{X} = \{X_0, X_1, X_2, X_3\}$.

## B.1 Computing $\phi$ for leaf nodes

Consider leaf node $X_3$. For $0 \leq m \leq \frac{\bar{r}}{1-\beta}$,

$$\phi_{X_3,0}(m) = \max\left\{m, \mathcal{P}\left(X_3 = 0 \mid X_2 = 0\right) \cdot \left[r(X_3, 0) + \beta\Phi_{\emptyset,0}(m)\right]\right.$$

$$\left. + \mathcal{P}\left(X_3 = 1 \mid X_2 = 0\right) \cdot \left[r(X_3, 1) + \beta\Phi_{\emptyset,1}(m)\right]\right\}$$

$$= \max\left\{m, \left(1 - \frac{1}{1 + e}\right) \cdot \left[r(X_3, 0) + \beta\Phi_{\emptyset,0}(m)\right] + \left(\frac{1}{1 + e}\right) \cdot \left[r(X_3, 1) + \beta\Phi_{\emptyset,1}(m)\right]\right\}$$
$$\text{(By Eq. (5) and Eq. (6))}$$

$$= \max\left\{m, \left(1 - \frac{1}{1 + e}\right) \cdot \left[0 + \beta m\right] + \left(\frac{1}{1 + e}\right) \cdot \left[1 + \beta m\right]\right\}$$
$$\text{(By definition of } r \text{ and since } \Phi_{\emptyset,0}(m) = \Phi_{\emptyset,1}(m) = m)$$

$$= \max\left\{m, \beta m + \frac{1}{1 + e}\right\}$$

In other words, we have

$$\phi_{X_3,0}(m) = \begin{cases} \beta m + \frac{1}{1+e} & \text{if } m \leq \frac{10}{1+e} \\ m & \text{if } m \geq \frac{10}{1+e} \end{cases}$$

By similar computations, we obtain

$$\phi_{X_3,1}(m) = \begin{cases} \beta m + \frac{e}{1+e} & \text{if } m \leq \frac{10e}{1+e} \\ m & \text{if } m \geq \frac{10e}{1+e} \end{cases}$$

Fig. 8 illustrates the piecewise functions $\phi_{X_3,0}$ and $\phi_{X_3,1}$. To represent these functions efficiently, we can track the changepoints and the linear pieces. For instance, the changepoint $\frac{10}{1+e}$ in $\phi_{X_3,0}$, the changepoint $\frac{10e}{1+e}$ in $\phi_{X_3,1}$, and the slope coefficient $a$ and intercept $b$ in the $y = am + b$ linear equation formula.

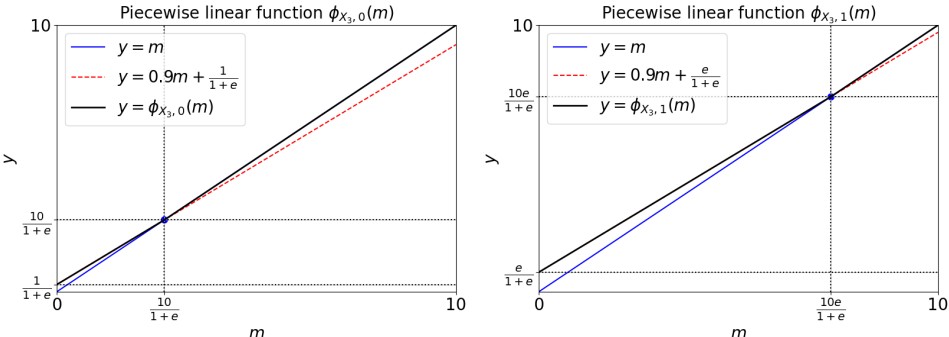

Figure 8: Illustration of the piecewise linear functions $\phi_{X_3,0}$ and $\phi_{X_3,1}$

By symmetry of $\mathcal{P}$ in our example, one can check that $\phi_{X_1,0} = \phi_{X_3,0}$ and $\phi_{X_1,1} = \phi_{X_3,1}$.

## B.2    Computing $\phi$ for non-leaf nodes

Now, let us consider the non-leaf node $X_2$. For $0 \le m \le \frac{\bar{r}}{1-\beta}$,

$$
\phi_{X_2,0}(m) = \max \left\{ m, \mathcal{P}\left(X_2 = 0 \mid X_0 = 0\right) \cdot \left[r(X_2, 0) + \beta\Phi_{\{X_3\},0}(m)\right] \right.
$$

$$
\left. + \mathcal{P}\left(X_2 = 1 \mid X_0 = 0\right) \cdot \left[r(X_2, 1) + \beta\Phi_{\{X_3\},1}(m)\right] \right\}
$$

$$
= \max \left\{ m, \left(1 - \frac{1}{1+e}\right) \cdot \left[0 + \beta\Phi_{\{X_3\},0}(m)\right] + \left(\frac{1}{1+e}\right) \cdot \left[1 + \beta\Phi_{\{X_3\},1}(m)\right] \right\}
$$

(By Eq. (5), Eq. (6), and definition of $r$)

Let us compute the function $\Phi_{\{X_3\},0}(m)$. Observe that

$$
\Phi_{\{X_3\},0}(m) = \frac{\bar{r}}{1-\beta} - \int_m^{\frac{\bar{r}}{1-\beta}} \prod_{Y \in \{X_3\}} \frac{\partial \phi_{Y,0}(k)}{\partial k} \, dk = 10 - \int_m^{10} \frac{\partial \phi_{X_3,0}(k)}{\partial k} \, dk
$$

Fig. 9 illustrates the piecewise linear function $\phi_{X_3,0}$ and corresponding piecewise constant function $\frac{\partial \phi_{X_3,0}(k)}{\partial k}$. Meanwhile, Fig. 10 shows a visualization of $\Phi_{\{X_3\},0}(m) = 10 - \int_m^{10} \frac{\partial \phi_{X_3,0}(k)}{\partial k} \, dk$.

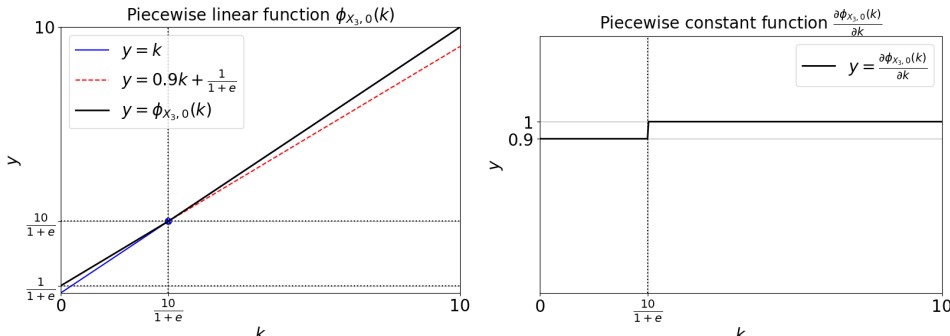

Figure 9: Illustration of the piecewise linear functions $\phi_{X_3,0}$ and $\frac{\partial \phi_{X_3,0}(k)}{\partial k}$

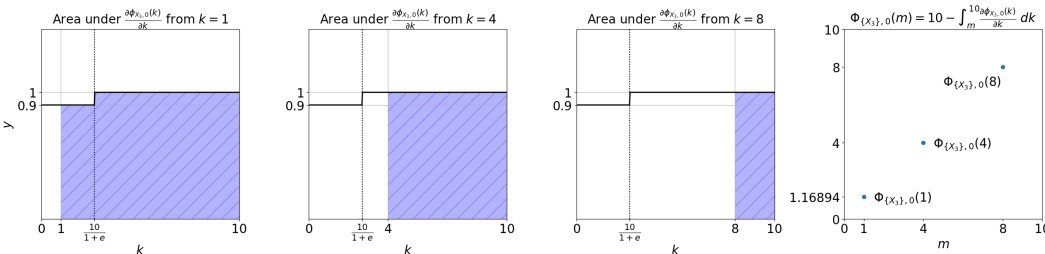

Figure 10: Visualization of $\Phi_{\{X_3\},0}(m) = 10 - \int_m^{10} \frac{\partial \phi_{X_3,0}(k)}{\partial k} \, dk$

In order to efficiently compute and represent $\Phi_{\{X_3\},0}(m)$, we turn to Proposition 5: $\Phi_{\{X_3\},0}(m) = \Phi_{\{X_3\},0}(0) + h(m)$, for some piecewise linear function $h$. Observe that $\Phi_{\{X_3\},0}(0)$ is just a constant while we showed in our proof that $h(m) = \int_m^{10} \frac{\partial \phi_{X_3,0}(k)}{\partial k} \, dk$. Intuitively, the idea is to first compute $h(m)$ by integrating $\frac{\partial \phi_{X_3,0}(k)}{\partial k}$ over the entire domain range of $0 \leq m \leq \frac{\bar{r}}{1-\beta}$, then "push the curves upwards" until the maximum value hits $\frac{\bar{r}}{1-\beta}$. For our example, one can compute $\Phi_{\{X_3\},0}(0) = (1 - \beta) \cdot \frac{10}{1+e}$ and derive that

$$\Phi_{\{X_3\},0}(m) = \begin{cases} \Phi_{\{X_3\},0}(0) + \beta m & \text{if } m \leq \frac{10}{1+e} \\ \Phi_{\{X_3\},0}(0) + m & \text{if } m \geq \frac{10}{1+e} \end{cases}$$

See Fig. 11 for an illustration.

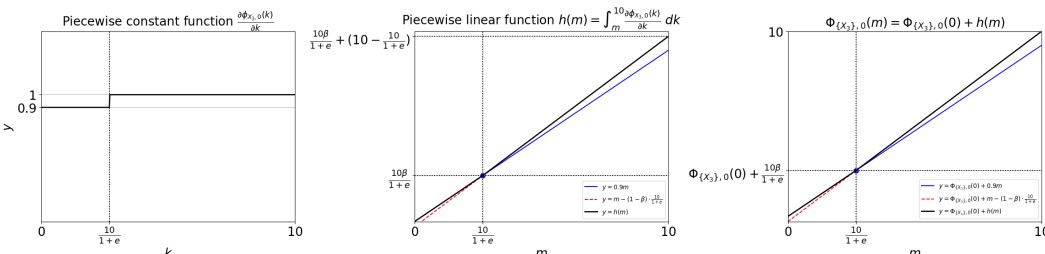

Figure 11: Illustration of how to apply Proposition 5 to compute $\Phi_{\{X_3\},0}(m)$

Using Proposition 5 in a similar manner for $\Phi_{\{X_3\},1}(m)$, we get $\Phi_{\{X_3\},1}(0) = (1 - \beta) \cdot \frac{10e}{1+e}$ and

$$\Phi_{\{X_3\},1}(m) = \begin{cases} \Phi_{\{X_3\},1}(0) + \beta m & \text{if } m \leq \frac{10e}{1+e} \\ \Phi_{\{X_3\},1}(0) + m & \text{if } m \geq \frac{10e}{1+e} \end{cases}$$

Continuing our calculations from above, we see that $\phi_{X_2,0}(m) = \max\{m, f_{X_2,0}(m)\}$, where

$$f_{X_2,0}(m) = \left(1 - \frac{1}{1+e}\right) \cdot \left[0 + \beta\Phi_{\{X_3\},0}(m)\right] + \left(\frac{1}{1+e}\right) \cdot \left[1 + \beta\Phi_{\{X_3\},1}(m)\right]$$

$$= \begin{cases} \beta^2 m + \beta(1-\beta)\frac{20e}{(1+e)^2} + \frac{1}{1+e} & \text{if } m \leq \frac{10}{1+e} \\ \frac{e\beta+\beta^2}{1+e}m + \beta(1-\beta)\frac{10e}{(1+e)^2} + \frac{1}{1+e} & \text{if } \frac{10}{1+e} \leq m \leq \frac{10e}{1+e} \\ \beta m + \frac{1}{1+e} & \text{if } m \geq \frac{10e}{1+e} \end{cases}$$

So,

$$\phi_{X_2,0}(m) = \max\{m, f_{X_2,0}(m)\}$$
$$= \begin{cases} \beta^2 m + \beta(1-\beta)\frac{20e}{(1+e)^2} + \frac{1}{1+e} & \text{if } m \leq \frac{10}{1+e} \\ \frac{e\beta+\beta^2}{1+e}m + \beta(1-\beta)\frac{10e}{(1+e)^2} + \frac{1}{1+e} & \text{if } \frac{10}{1+e} \leq m \leq \frac{\beta(1-\beta)\frac{10e}{(1+e)^2}+\frac{1}{1+e}}{1-\frac{e\beta+\beta^2}{1+e}} \\ m & \text{if } m \geq \frac{\beta(1-\beta)\frac{10e}{(1+e)^2}+\frac{1}{1+e}}{1-\frac{e\beta+\beta^2}{1+e}} \end{cases}$$

The piecewise function $\phi_{X_2,1}(m)$ can be computed in the same way.

$$\phi_{X_2,1}(m) = \begin{cases} \beta^2 m + \beta(1-\beta)\frac{10(1+e^2)}{(1+e)^2} + \frac{e}{1+e} & \text{if } m \leq \frac{10}{1+e} \\ \frac{\beta+e\beta^2}{1+e}m + \beta(1-\beta)\frac{10e^2}{(1+e)^2} + \frac{e}{1+e} & \text{if } \frac{10}{1+e} \leq m \leq \frac{\beta(1-\beta)\frac{10e^2}{(1+e)^2} + \frac{e}{1+e}}{1-\beta\frac{1+e\beta}{1+e}} \\ m & \text{if } m \geq \frac{\beta(1-\beta)\frac{10e^2}{(1+e)^2} + \frac{e}{1+e}}{1-\beta\frac{1+e\beta}{1+e}} \end{cases}$$

Finally, for labels $b \in \boldsymbol{\Omega}$, we compute $\phi_{X_0,b}$ in a similar fashion, with the only non-trivial computation being $\prod_{Y \in \text{Ch}(X_0)} \frac{\partial \phi_{Y,v}(k)}{\partial k} \, dk$, where $\text{Ch}(X_0) = \{X_1, X_2\}$ and $v \in \boldsymbol{\Omega}$. Here, we take the product of the corresponding piecewise constant functions before integrating it to obtain the corresponding piecewise linear function $h(m)$.

## B.3 Bounding the number of pieces

Throughout our computation, we relied heavily on the fact that the recursive functions $\phi$ and $\Phi$ can be computed by manipulating piecewise linear functions. Here, we show that the number of pieces we need to manipulate scales reasonably with the size of the input rooted forest $\mathcal{G} = (\mathbf{X}, \mathbf{E})$. A key part of the proof relies on the following observation:

**Observation 7.** Multiplying by constants, adding constants, differentiation, and integration do not affect the number of changepoints in piecewise linear functions.

Towards a formal proof, let us define some additional notation. For any node $X \in \mathbf{X}$, let $\mathbf{T}_X$ denote the subtree rooted at $X$ and $|\mathbf{T}_X|$ denote the number of nodes in this subtree, *excluding $X$ itself*. Meanwhile, for any piecewise linear function $f$, we write $c(f)$ and $\#(f)$ to denote the *set* and *number* of changepoints required to represent $f$ respectively, i.e., $\#(f) = |c(f)|$. That is, $\#$ counts the number of cases in the function representation, minus 1. For instance, $\#(\phi_{X,b}) \leq 1$ for any leaf node $X \in \mathbf{X}$ due to the maximization against the linear function $y = m$.

**Lemma 8.** *For any arbitrary node $X \in \mathbf{X}$ and label $b \in \boldsymbol{\Omega}$, we have $\#(\phi_{X,b}) \leq 1 + |\boldsymbol{\Omega}| \cdot |\mathbf{T}_X|$.*

Observe that the upper bound of Lemma 8 is *independent* of the actual label $b \in \boldsymbol{\Omega}$. The intuition behind this independence can been seen in Eq. (1) where $b$ only affects a multiplicative scaling via the conditional distribution value, which by itself does not affect the number of changepoints; see Observation 7. An important implication of Lemma 8 is that the maximum number of changepoints we ever need to manipulate for a rooted tree rooted at $X_{\text{root}}$ in $\mathcal{G}$ is at most $|\boldsymbol{\Omega}| \cdot |\mathbf{T}_{X_{\text{root}}}|$.

*Proof of Lemma 8.* We prove the claim by induction over the tree structure, starting from the leaves.

**Base case ($X$ is a leaf node):** Then, $\text{Ch}(X) = \emptyset$ and $\#(\phi_{X,b}) \leq 1$.

**Inductive case ($X$ is a non-leaf node):** Recall Eq. (1) and Eq. (2) for $\text{Ch}(X) \neq \emptyset$:

$$\phi_{X,b}(m) = \max\left\{m, \sum_{v \in \boldsymbol{\Omega}} \mathcal{P}(X = v \mid \text{Pa}(X) = b) \cdot \left[r(X,v) + \beta\Phi_{\text{Ch}(X),v}(m)\right]\right\}$$

$$\Phi_{\text{Ch}(X),v}(m) = \frac{\bar{r}}{1-\beta} - \int_m^{\frac{\bar{r}}{1-\beta}} \prod_{Y \in \text{Ch}(X)} \frac{\partial \phi_{Y,v}(k)}{\partial k} \, dk$$

Defining $g_{X,b,v}(m) = \mathcal{P}(X = v \mid \text{Pa}(X) = b) \cdot \left[r(X,v) + \beta\Phi_{\text{Ch}(X),v}(m)\right]$, we can rewrite $\phi_{X,b}(m)$ as $\phi_{X,b}(m) = \max\{m, \sum_{v \in \boldsymbol{\Omega}} g_{X,b,v}(m)\}$. For any two distinct labels $b, b' \in \boldsymbol{\Omega}$, Observation 7 tells us that the set of changepoints in $\phi_{X,b}$ and $\phi_{X,b'}$ differ by at most one, depending on depending on where the intersection $y = m$ occurs for each label. That is,

$$|s(\phi_{X,b}) \setminus s(\phi_{X,b'})| \leq 1 \tag{7}$$

So,

$$\#(\phi_{X,b}) = \#\left(\max\left\{m, \sum_{v\in\mathbf{\Omega}} g_{X,b,v}\right\}\right) \qquad \text{(By Eq. (1))}$$

$$\leq 1 + \#\left(\sum_{v\in\mathbf{\Omega}} g_{X,b,v}\right) \qquad \text{(Due to maximization with } y = m)$$

$$= 1 + \bigcup_{v\in\mathbf{\Omega}} \#\left(g_{X,b,v}\right)$$

$$= 1 + \bigcup_{v\in\mathbf{\Omega}} \#\left(\Phi_{\mathrm{Ch}(X),v}\right) \qquad \text{(By Observation 7)}$$

$$= 1 + \bigcup_{v\in\mathbf{\Omega}} \bigcup_{Y\in\mathrm{Ch}(X)} \#\left(\phi_{Y,v}\right) \qquad \text{(By Eq. (2))}$$

$$= 1 + \bigcup_{Y\in\mathrm{Ch}(X)} \bigcup_{v\in\mathbf{\Omega}} \#(\phi_{Y,v}) \qquad \text{(Switching the ordering of the unions)}$$

$$\leq 1 + \bigcup_{Y\in\mathrm{Ch}(X)} \left(|\mathbf{\Omega}| - 1 + \max_{v\in\mathbf{\Omega}} \#(\phi_{Y,v})\right) \qquad \text{(Applying Eq. (7) over all } |\mathbf{\Omega}| \text{ labels)}$$

$$\leq 1 + \sum_{Y\in\mathrm{Ch}(X)} \left(|\mathbf{\Omega}| - 1 + (1 + |\mathbf{\Omega}| \cdot |\mathbf{T}_Y|)\right)$$

$$\qquad \text{(By induction hypothesis and replacing } \bigcup \text{ with } \sum)$$

$$= 1 + |\mathbf{\Omega}| \cdot \sum_{Y\in\mathrm{Ch}(X)} (1 + |\mathbf{T}_Y|) \qquad \text{(Rearranging)}$$

$$= 1 + |\mathbf{\Omega}| \cdot |\mathbf{T}_X| \qquad \text{(By definition of } |\mathbf{T}_X| \text{ and } |\mathbf{T}_Y|)$$

$$\square$$

## C  Application to network-based disease testing

In this section, we explore the application of network-based disease testing motivated in Section 1 where the goal is to identify infected individuals given knowledge of their interaction network; see Fig. 1 for an illustration. Here, each node represents an individual with a binary infection status (infected or not), and edges represent sexual interactions. Frontier testing is a natural operational constraint: test outcomes significantly influence beliefs about neighboring individuals, making it efficient to expand testing along the observed frontier.

Building upon our notations in Section 2, a joint distribution $\mathcal{P}_\theta$ parameterized by $\theta$ is written as $\mathcal{P}_\theta(\mathbf{x}) = \mathcal{P}_\theta(X_1 = x_1, \ldots, X_n = x_n)$ for $\mathbf{x} \in \mathbf{\Omega}^n$.

### C.1  A MRF-based joint model of infection status

We model the joint distribution over $n$ individuals' infection statuses using a pairwise MRF defined over the interaction graph $\mathcal{G} = (\mathbf{X}, \mathbf{E})$, where each node $X_i$ represents an individual with a binary latent variable $X_i \in \{0,1\}$ indicating its HIV status, and each edge $\{X_i, X_j\} \in \mathbf{E}$ indicates a reported sexual interaction. Each individual also has associated covariates $\mathbf{c}^{(i)} \in \mathbb{R}^d$ and the joint distribution over all statuses $\mathbf{X} = (X_1, \ldots, X_n)$ is defined in terms of unary and pairwise potential functions $\phi_i(x_i)$ for each individual $i$, and $\phi_{i,j}(x_i, x_j)$ for each edge $\{i,j\} \in \mathbf{E}$, with $1 \leq i < j \leq n$[11]:

$$\phi_i(x_i) = \exp\left(\theta_1^\top f_1(x_i, \mathbf{c}^{(i)})\right) \quad \text{and} \quad \phi_{i,j}(x_i, x_j) = \exp\left(\theta_2^\top f_2(x_i, x_j, \mathbf{c}^{(i)}, \mathbf{c}^{(j)})\right)$$

for some feature mapping functions $f_1 : \{0,1\} \times \mathbb{R}^d \to \mathbb{R}^{2+2d}$ and $f_2 : \{0,1\}^2 \times \mathbb{R}^{2d} \to \mathbb{R}^{4+5d}$, and parameters $\theta_1 \in \mathbb{R}^{2+2d}$ and $\theta_2 \in \mathbb{R}^{4+5d}$. We adopt the maximum entropy principle [Jay57,

---

[11]For notational simplicity, we write $f_2(x_j, x_i, \mathbf{c}^{(j)}, \mathbf{c}^{(i)})$ to mean $f_2(x_i, x_j, \mathbf{c}^{(i)}, \mathbf{c}^{(j)})$ for any $i < j$.

WJ08, Wu12] to parameterize these factors, with feature maps $f_1$ and $f_2$ defined as monomials up to quadratic terms of the covariate variables while respecting symmetry:

$$f_1(x_i, \mathbf{c}^{(i)}) = \left(1, x_i, \mathbf{c}_1^{(i)}, x_i \mathbf{c}_1^{(i)}, \ldots, \mathbf{c}_d^{(i)}, x_i \mathbf{c}_d^{(i)}\right)^\top \in \mathbb{R}^{2+2d} \tag{8}$$

$$f_2(x_i, x_j, \mathbf{c}^{(i)}, \mathbf{c}^{(j)}) = \Big(1, x_i x_j, (1-x_i)x_j + x_i(1-x_j), (1-x_i)(1-x_j),$$

$$\mathbf{v}(x_i, x_j, \mathbf{c}_1^{(i)}, \mathbf{c}_1^{(j)}), \ldots, \mathbf{v}(x_i, x_j, \mathbf{c}_d^{(i)}, \mathbf{c}_d^{(j)})\Big)^\top \in \mathbb{R}^{4+5d} \tag{9}$$

where $\mathbf{v} : \{0,1\}^2 \times \mathbb{R}^2 \to \mathbb{R}^5$ is defined as

$$\mathbf{v}(a,b,c,d) = \Big(c+d, ab(c+d), a(1-b)c+(1-a)bd, (1-a)bc+a(1-b)d, (1-a)(1-b)(c+d)\Big)$$

The joint probability is then:

$$\mathcal{P}_{\theta_1,\theta_2}(\mathbf{X} = \mathbf{x}) = \frac{1}{z(\theta_1,\theta_2)} \exp\left(\sum_{i=1}^n \theta_1^\top f_1(x_i, \mathbf{c}_i) + \sum_{\{i,j\} \in \mathbf{E}} \theta_2^\top f_2(x_i, x_j, \mathbf{c}_i, \mathbf{c}_j)\right) \tag{10}$$

where $z(\theta_1, \theta_2)$ is the partition function. This formulation encodes both individual risk via covariates and dependency via pairwise interactions, capturing correlation in infection status across the network. To reduce model complexity and reflect data limitations, we use parameter sharing, i.e. all unary (resp. pairwise) factors share the same parameters $\theta_1$ (resp. $\theta_2$). Although exact inference in general MRFs is intractable, the sexual contact networks we consider are typically sparse, bipartite, or even tree-structured — as in contact tracing studies — where efficient inference algorithms apply [KF09]. Therefore, we assume access to an inference oracle, and focus on the primary challenge: adaptive sequential testing under frontier constraints.

At each time step $t = \{1, 2, \ldots, n\}$, we select an untested individual to test on the frontier and observe their HIV status. Recalling the definition of AFEG (Definition 1), we can model the interaction network as $\mathcal{G}$, and each test as acting on a node in $\mathcal{G}$. The reward function for testing individual $X$ and revealing status $b \in \{0, 1\}$ is simply $r(X, b) = b$, and any discount factor $\beta \in (0, 1)$ would encourage identifying HIV+ individuals as early as possible. Importantly, discounting reflects both practical constraints – such as sudden funding cuts [UNA25] – and clinical importance of early diagnosis, which improves patient outcomes and limits transmission [CCM+11]. See also [RN21] for other natural justifications for using discount factors $\beta$ in modeling long-term policy rewards. Finally, to apply the infinite horizon framework of AFEG in our finite testing setting, we simply zero subsequent rewards after every individual has been already tested.

## C.2 Learning the distributional parameters from data

To apply our model to a new population with unknown HIV statuses, we must first estimate the parameters of the joint distribution described in Appendix C.1. We assume access to a historical dataset in which both the covariates and true HIV statuses are known. Classical approaches to MRF parameter learning such as [AKN06] typically assume access to multiple independent samples drawn from a fixed graphical model. Unfortunately, in our case, we only have access to only *a single observed realization* of infection statuses in our past data. This means that the maximum likelihood estimation (MLE) distribution $\mathcal{P}$ that describes the dataset is simply the degenerate point distribution that places full probability mass on the single realization.

To learn a meaningful but non-degenerate transmission probabilities, we consider an intuitive way to model the joint probabilities based on a factor graph induced by the input graph structure. More specifically, we define unary factor potentials for each individual node and pairwise factor potentials for each edge present in the graph, governed by global parameters $\theta_1$ and $\theta_2$ respectively. The hope is that this simple formulation serves as a regularization allows us to recover meaningful disease-specific parameters so that we can define joint distributions $\mathcal{P}$ on new interaction graphs for the same disease.

We adopt a maximum likelihood estimation (MLE) approach to learn these $\theta_1$ and $\theta_2$ parameters with respect to this single realization under the MRF model: $\theta^* = \arg\max_{\theta_1,\theta_2} \log \mathcal{P}(\mathbf{x}; \theta_1, \theta_2)$. However,

exact MLE is intractable in general due to the partition function $z(\theta_1, \theta_2)$, whose computation requires summing over all $2^n$ configurations of node labels. To sidestep this difficulty, we instead optimize the pseudo-likelihood [Bes75], which approximates the joint likelihood by the product of conditional distributions for each node given its neighbors:

$$\widetilde{\mathcal{P}}_{\theta_1,\theta_2}(\mathbf{x}) = \prod_{i=1}^{n} \mathcal{P}_{\theta_1,\theta_2}(X_i = x_i \mid \mathbf{x}_{-i}) = \prod_{i=1}^{n} \frac{\mathcal{P}_{\theta_1,\theta_2}(\mathbf{x})}{\mathcal{P}_{\theta_1,\theta_2}(X_i = 0, \mathbf{x}_{-i}) + \mathcal{P}_{\theta_1,\theta_2}(X_i = 1, \mathbf{x}_{-i})} \quad (11)$$

This objective is tractable and differentiable with respect to $\theta_1$ and $\theta_2$, and can be efficiently optimized using gradient-based methods. Once learned, these parameters can be used to define a new factor graph for any unseen population with known covariates and network structure, thereby guiding the adaptive testing policy.

In general, a closed-form solution for the maximum pseudolikelihood estimator is unlikely to exist due to the nonlinear dependence of the local conditional distributions on the parameters. However, as the following lemma shows, we can derive closed-form gradients and rely on gradient-based optimization methods to compute parameter estimates for $\theta_1$ and $\theta_2$.

**Lemma 9.** *Let $\mathcal{G} = (\mathbf{X}, \mathbf{E})$ be a graph over $n$ nodes, where each node $X_i \in \mathbf{X}$ has binary label $x_i \in \{0, 1\}$, covariates $\mathbf{c}_i \in \mathbb{R}^d$, and neighborhood $\mathbf{N}(X_i)$. Define feature maps $f_1(x_i, \mathbf{c}_i)$ and $f_2(x_i, x_j, \mathbf{c}_i, \mathbf{c}_j)$ as per Eq. (8) and Eq. (9) respectively, shared parameters $\theta_1 \in \mathbb{R}^{2+2d}$ and $\theta_2 \in \mathbb{R}^{4+5d}$, and the joint probability as per Eq. (10). Then, the log-pseudolikelihood gradients are:*

$$\frac{\partial \log \widetilde{\mathcal{P}}_{\theta_1,\theta_2}(\mathbf{x})}{\partial \theta_1} = \sum_{i=1}^{n} \alpha_i \cdot \left( f_1(1, \mathbf{c}^{(i)}) - f_1(0, \mathbf{c}^{(i)}) \right)$$

$$\frac{\partial \log \widetilde{\mathcal{P}}_{\theta_1,\theta_2}(\mathbf{x})}{\partial \theta_2} = \sum_{i=1}^{n} \alpha_i \cdot \sum_{X_j \in \mathbf{N}(X_i)} \left( f_2(1, x_j, \mathbf{c}^{(i)}, \mathbf{c}^{(j)}) - f_2(0, x_j, \mathbf{c}^{(i)}, \mathbf{c}^{(j)}) \right)$$

*where the common coefficient $\alpha_i = x_i - \mathcal{P}_{\theta_1,\theta_2}(X_i = 1 \mid \mathbf{x}_{-i})$ can be computed efficiently without computing $z(\theta_1, \theta_2)$ for all $i \in [n]$.*

The full proof of Lemma 9 is given in Appendix F.2. Note that parameter fitting may not necessary recover the exact dynamics of the underlying real-world problem in general. That is, $\mathcal{P}_{\widehat{\theta}_1,\widehat{\theta}_2} \neq \mathcal{P}$ in general, where $\widehat{\theta}_1$ and $\widehat{\theta}_2$ are produced parameter estimates assuming the the MRF model defined in Appendix C.1 and $\mathcal{P}$ is the true unknown underlying joint probabilities (which may even lie outside the model class defined by Appendix C.1). As such, we also provide an error analysis in Appendix F.3 to bound the loss of the attainable discounted accumulated reward of an optimal policy computed on $\mathcal{P}_{\widehat{\theta}_1,\widehat{\theta}_2}$ while being executing on $\mathcal{P}$.

# D   Further discussions

## D.1   Broader impact

A central motivation of this work is the urgent need for resource-efficient strategies in global public health. In the face of constrained resources and diminishing funding for disease control programs [UNA25], optimizing the allocation of testing efforts is increasingly critical. Our framework enables targeted, adaptive exploration of interaction networks, and is particularly well-suited to settings where prior data can inform transmission structure through a learned distribution $\mathcal{P}$, possibly handcrafted by domain experts. This balance between data-driven structure and real-time adaptivity makes the AFEG framework a compelling tool for improving public health decision-making.

## D.2   Limitations

While our framework makes several modeling assumptions to enable tractable inference and principled decision making, these also define the scope within which our results apply. Each of these extensions below presents a well-motivated and technically rich research challenge building on the foundation we establish.

First, we assume that the interaction graph $\mathcal{G}$ is known and fixed. This is a reasonable assumption in many structured public health applications, such as contact tracing or intervention planning, where

the network is elicited or constructed from prior data. However, generalizing our methods to handle uncertain or dynamically evolving graphs is a promising direction for future work.

Second, our paper is focused on $\mathcal{P}$ oracles. One way to model this is via MRFs: in Appendix C.1, we discuss how to model network-based disease testing using pairwise Markov Random Fields with shared parameters, which provide a flexible yet interpretable model class for encoding local dependencies in infection status. A more complicated model could be designed and used based on further inputs from domain experts. It is also worth noting that while exact inference of MRFs is tractable on sparse graphs (as motivated by real-world sexual networks), scaling to larger or denser networks may require approximate inference or amortized learning approaches. One can also consider relaxing to Gaussian random fields [ZLG03], as per the literature of active searching on graphs; see Appendix A.

Third, our theoretical guarantees for the GITTINS policy are currently limited to tree-structured graphs; nonetheless, our empirical results demonstrate that it performs competitively even on graphs with low to moderate treewidth.

Finally, while exact inference is tractable on sparse graphs (as motivated by real-world sexual networks), scaling to larger or denser networks may require approximate inference or amortized learning approaches.

### D.3 Fairness considerations

A notable advantage of our AFEG framework is its flexibility to accommodate fairness constraints or objectives in sequential decision making. Since our model maintains posterior beliefs over individual infection risks via a probabilistic graphical model, fairness-aware modifications can be naturally incorporated at the policy level. For example, one can enforce demographic parity by requiring individuals from different subpopulations to have equal testing probabilities over time, or impose group-specific constraints on exposure or false negative rates. More generally, fairness can be encoded through soft constraints or regularization terms in the policy objective, or via hard constraints within the action-selection mechanism. Notably, fairness interventions can also be incorporated directly through the reward function. To prioritize historically underserved groups, one could upweight successful identifications among protected populations — for instance, by defining $r(X, b) = b \cdot \alpha \cdot \mathbb{I}_{\text{protected}}$ for some $\alpha > 1$. Since GITTINS policies depend only on the reward structure and not on group identity per se, such node-dependent reward shaping modifications preserve optimality guarantees on trees and maintain empirical performance on general graphs. Furthermore, our Bayesian formulation allows dynamic reweighting or calibration as more data is revealed, enabling adaptive policies that balance efficiency and equity. Exploring how to systematically integrate such fairness interventions into frontier-constrained graph exploration is a promising direction for future work, particularly in public health settings where equitable resource allocation is essential.

## E   Experimental details and more experimental results

While we do not release the ICPSR dataset, in accordance with its terms of use, interested researchers may independently access it via https://www.icpsr.umich.edu/web/ICPSR/studies/22140. To facilitate reproducibility, our experimental scripts and the code used for data preprocessing and parameter estimation are available on our Github repository[12].

### E.1   Defining joint probability distribution $\mathcal{P}$ in our experiments

As described in Section 4, all of our experiments are modeled after the network-based disease testing application discussed in Section 1.1, where node labels are binary and a reward of 1 is received if and only if the revealed label indicates a positive diagnosis. Accordingly, the joint distribution $\mathcal{P}$ over node labels follows the pairwise MRF formulation described in Appendix C.1. Since the covariates in our real-world dataset [MR11][13] are categorical, we apply one-hot encoding to transform them into binary vectors. For consistency, our synthetic experiments also use binary covariates.

---

[12]https://github.com/cxjdavin/adaptive-frontier-exploration-on-graphs
[13]This de-identified, public-use dataset is available at https://www.icpsr.umich.edu/web/ICPSR/studies/22140 under ICPSR's terms of use. No IRB approval was required for our use of this data.

**Synthetic experiments in Section 4.1 and Section 4.2.** For each instance, we sample random parameters $\theta_1 \in \mathbb{R}^{2+2d}$ and $\theta_2 \in \mathbb{R}^{4+5d}$, and assign each node a random binary covariate vector of dimension $d$. The joint distribution $\mathcal{P}$ is then defined according to Eq. (10). Since all policies are agnostic to the choice of $d$, we fix $d = 5$ to keep computation time manageable.

**Constructing $\mathcal{P}$ from the real-world dataset in Section 4.3.** The real-world dataset comprises a collection of network-based surveys across eight studies, each recording disease statuses for five sexually transmitted infections (Gonorrhea, Chlamydia, Syphilis, HIV, and Hepatitis). We first filtered the data to retain only edges denoting reported sexual interactions and excluded individuals with missing or ambiguous disease status labels. Because the covariates are shared across diseases but transmission dynamics differ, we aggregated data across studies and split it by disease. See Table 1 for dataset summary statistics. For covariates, we used categorical survey responses capturing demographic and behavioral factors relevant to disease transmission such as gender[14], homelessness, sex work involvement, etc. A total of 17 categorical variables were one-hot encoded into binary vectors of dimension $d = 72$. We then applied the parameter fitting procedure from Appendix F.2 to estimate $\theta_1 \in \mathbb{R}^{2+2d}$ and $\theta_2 \in \mathbb{R}^{4+5d}$ for each disease-specific dataset, and constructed the corresponding pairwise MRF $\mathcal{P}$ using Eq. (10).

Table 1: Summary statistics of real-world sexual interaction graphs [MR11]. Approximate treewidth is obtained by computing a tree decomposition using `networkx`'s `treewidth_min_fill_in`. The graphs for Gonorrhea and Chlamydia are identical but the infection rates are different, i.e., not everyone is infected with both diseases.

| Sexually transmitted disease | Gonorrhea | Chlamydia | Syphilis | HIV | Hepatitis |
|---|---|---|---|---|---|
| Number of infected | 66 | 963 | 44 | 88 | 117 |
| Number of individuals | 2079 | 2079 | 542 | 778 | 1732 |
| Number of edges | 1326 | 1326 | 519 | 793 | 1260 |
| Diameter of graph | 8 | 8 | 12 | 13 | 24 |
| Approximate tree width | 1 | 1 | 16 | 16 | 6 |

## E.2    Full results for Section 4.1

As described in Section 4.1, we evaluated the policies against a familly of randomly generated synthetic trees on $n \in \{10, 50, 100\}$ nodes across various discount facotrs $\beta \in \{0.5, 0.7, 0.9\}$. Fig. 3 in Section 4.1 shows only the figures with discount factor $\beta = 0.9$. See Fig. 12 for the full $3 \times 3$ plot.

## E.3    Full results for Section 4.2

As described in Section 4.2, we progressively add random non-tree edges to the synthetic trees to observe the change in relative performance of our policies. Across all experiments, we consider a discount factor of $\beta = 0.9$ and add $\{0, 2, 4, 6, 8, 10\}$ extra edges to each graph. Fig. 4 in Section 4.2 shows only the figures for $n = 50$; see Fig. 13 for results on $n \in \{10, 100\}$. While 10 edges may seem like a small number, recall that trees have $n - 1$ edges. So, adding 10 edges correspond to adding roughly $100\%$, $20\%$, and $10\%$ adding additional edges to each graph for $n \in \{10, 50, 100\}$ respectively.

## E.4    Full results for Section 4.3

Table 2 provides the summary statistics of the subsampled real-world graphs which our experiments in Fig. 5 are based on.

In the main paper, we showed empirical results for when policies only have access to a noisy version $\mathcal{Q}_\varepsilon$ of the true underlying distribution $\mathcal{P}$ for the HIV dataset. Fig. 14 shows the results for all 5 diseases.

---

[14]See [FHA+98, Scu18] for evidence on gender differences in HIV susceptibility.

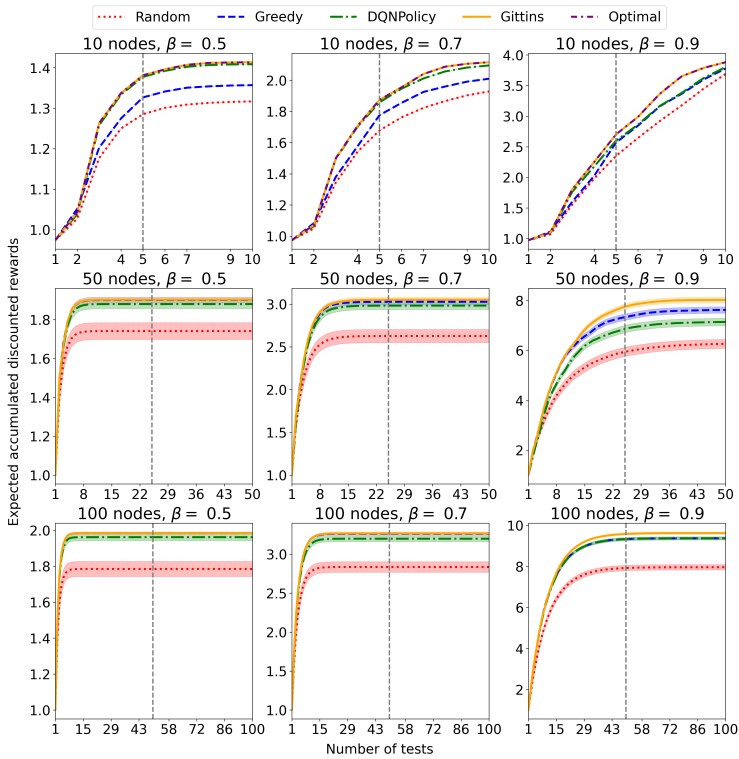

Figure 12: Full experimental results for synthetic tree experiments.

Table 2: Summary statistics of subsampled real-world sexual interaction graphs from [MR11]. CC stands for connected component, Max. depth refers to the maximum BFS tree depth for GITTINS, and approximate treewidth is obtained by using networkx's treewidth_min_fill_in.

| Disease | # Nodes | # Edges | Forest? | Diameter | # CC | Max. depth | Apx. treewidth |
|---|---|---|---|---|---|---|---|
| Gonorrhea | 300 | 195 | ✓ | 5 | 105 | 3 | 1 |
| Chlamydia | 300 | 195 | ✓ | 5 | 105 | 4 | 1 |
| Syphilis | 433 | 456 | ✗ | 12 | 97 | 9 | 16 |
| HIV | 305 | 390 | ✗ | 12 | 39 | 8 | 16 |
| Hepatitis | 300 | 184 | ✗ | 5 | 125 | 5 | 2 |

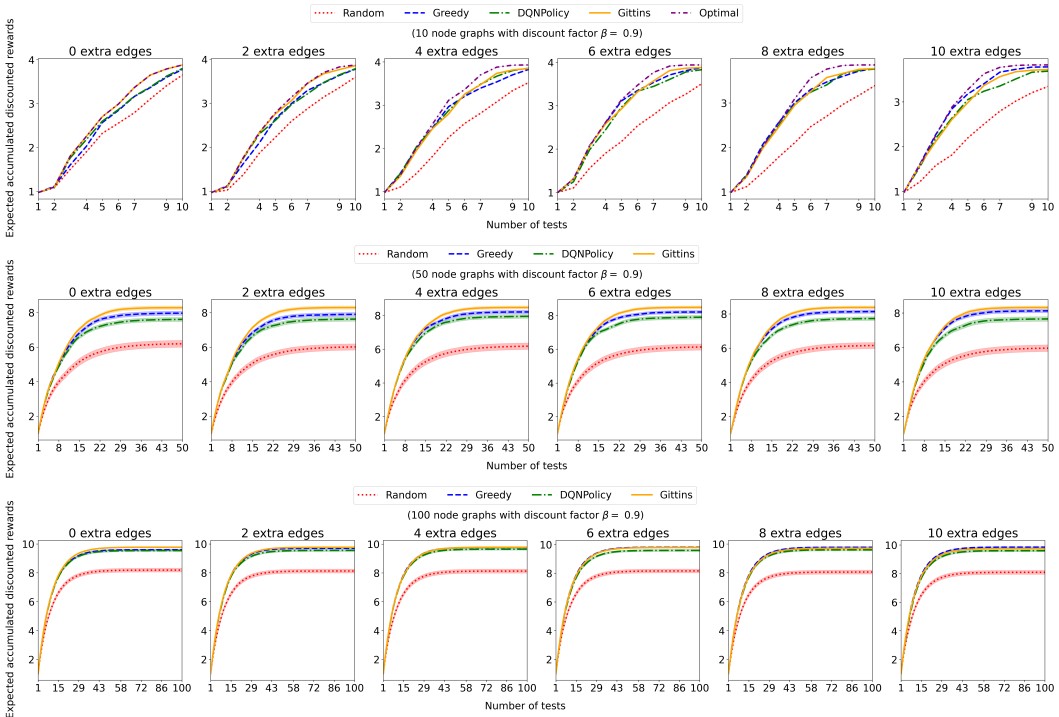

Figure 13: Full experimental results for synthetic non-tree experiments. Observe that the relative performance gains of GITTINS over other baselines decreases as we deviate from a tree. Furthermore, in the small $n = 10$ (top row) instances where we can run OPTIMAL, we see that GITTINS is no longer optimal as more non-tree edges are added, as expected.

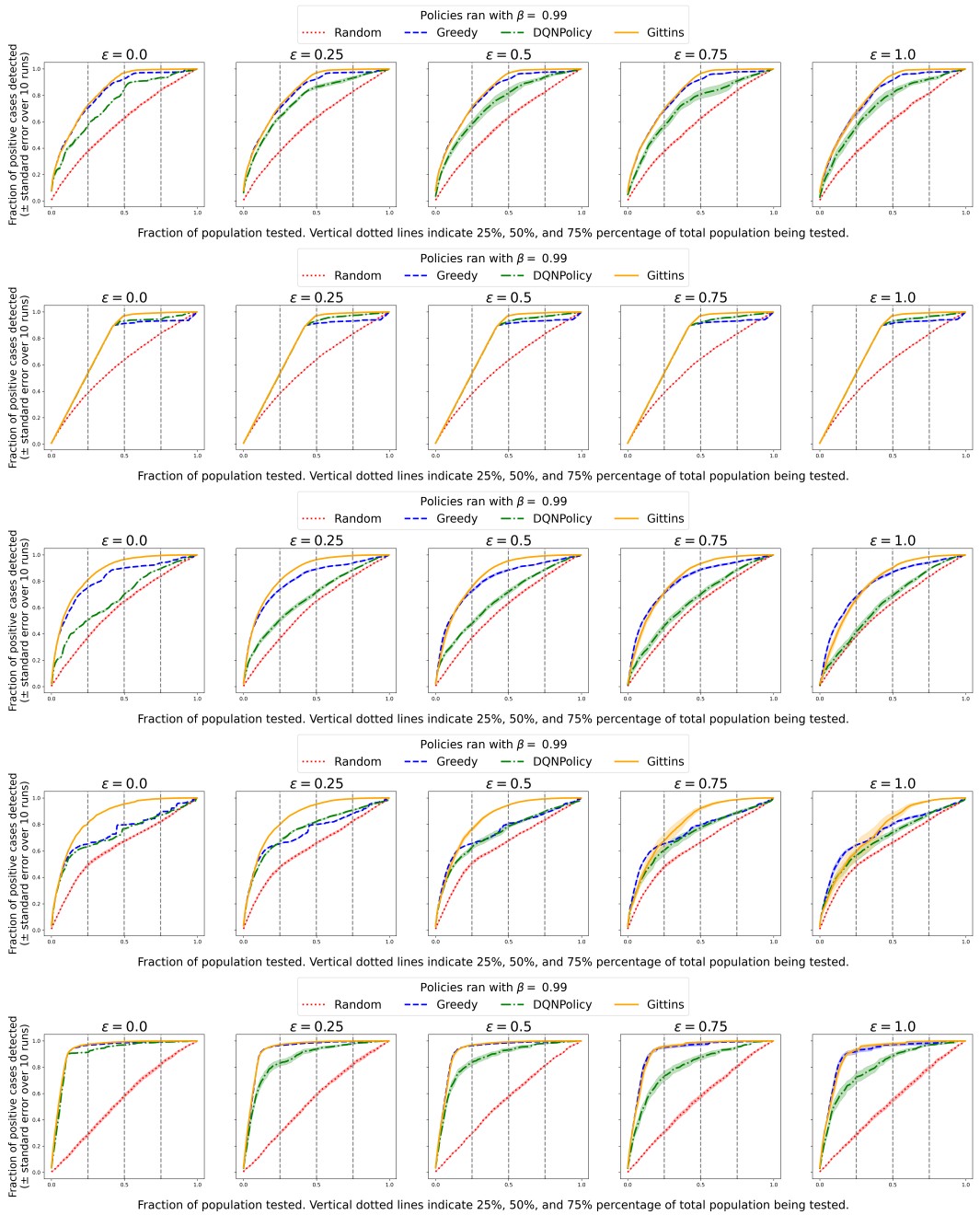

Figure 14: Experiments on the datasets of all 5 diseases where policies only have access noisy version $\mathcal{Q}_\varepsilon$ of the underlying distribution $\mathcal{P}$ on the HIV dataset. Error bars illustrate the standard error due to 10 random instantiations of $\mathcal{Q}_\varepsilon$ for each corresponding value of $\varepsilon$.

# F   Deferred proof details

## F.1   Gittins proofs

**Lemma 4.** *For any node $X \in \mathbf{X}$ and label $b \in \Omega$, $\phi_{X,b}(m)$ is a non-decreasing piecewise linear function over $m \in [0, \bar{r}/(1 - \beta)]$.*

*Proof.* Let us induct on the nodes from the leaf towards the root while recalling definitions of $\phi$ and $\Phi$ from Eq. (1) and Eq. (2) respectively.

**Base case ($X$ is a leaf):** For any $m \in [0, \frac{\bar{r}}{1-\beta}]$, we have $\Phi_{\mathrm{Ch}(X),b}(m) = \Phi_{\emptyset,b}(m) = m$, and so

$$\phi_{X,b}(m) = \max \left\{ m, \sum_{v \in \mathbf{\Omega}} \mathcal{P}(X = v \mid \mathrm{Pa}(X) = b) \cdot \left[ r(v) + \beta \cdot \Phi_{\mathrm{Ch}(X),v}(m) \right] \right\}$$

$$= \max \left\{ m, \sum_{v \in \mathbf{\Omega}} \mathcal{P}(X = v \mid \mathrm{Pa}(X) = b) \cdot \left[ r(v) + \beta m \right] \right\}$$

$$= \max \left\{ m, \beta m + \sum_{v \in \mathbf{\Omega}} \mathcal{P}(X = v \mid \mathrm{Pa}(X) = b) \cdot r(v) \right\}$$

Since $\sum_{v \in \mathbf{\Omega}} \mathcal{P}(X = v \mid \mathrm{Pa}(X) = b) \cdot r(v)$ is a constant with respect to $m$, we see that $\phi_{X,b}(m)$ is non-decreasing with respect to $m$. Furthermore, since $m$ and $\beta m + \sum_{v \in \mathbf{\Omega}} \mathcal{P}(X = v \mid \mathrm{Pa}(X) = b) \cdot r(v)$ are both linear functions in $m$, combining them via the max operator into $\phi_{X,b}(m)$ yields a piecewise linear function of $m$ with at most 2 pieces.

**Inductive case ($X$ is a not a leaf):** Set $\mathbf{S} = \mathrm{Ch}(X)$ in Eq. (2). Consider $\frac{\partial \phi_{Y,b}(k)}{\partial k}$ for an arbitrary $Y \in \mathrm{Ch}(X)$ and label $b \in \Omega$. By induction hypothesis, we know that $\phi_{Y,b}(k)$ is a piecewise linear function in $k$, and so $\frac{\partial \phi_{Y,b}(k)}{\partial k}$ is a piecewise *constant* function in $k$. Thus, the product of $\prod_{Y \in \mathrm{Ch}(X)} \frac{\partial \phi_{Y,b}(k)}{\partial k}$ is a piecewise *constant* function, and the integral $\int_m^{\frac{\bar{r}}{1-\beta}} \prod_{Y \in \mathrm{Ch}(X)} \frac{\partial \phi_{Y,b}(k)}{\partial k} \, dk$ is a piecewise linear function of $m$. Therefore, since $\frac{\bar{r}}{1-\beta}$ is a constant with respect to $m$, we have that $\Phi_{\mathrm{Ch}(X),b}(m)$ is a piecewise linear function of $m$. Finally, similar to the base case argument above, we see that $\phi_{X,b}(m)$ is non-decreasing with respect to $m$ and $\phi_{X,b}(m)$ is a piecewise linear function of $m$ because the $\mathcal{P}(X = v \mid \mathrm{Pa}(X) = b)$ and $r(v)$ terms are constants with respect to $m$. As a remark, if $X$ is the root, we simply replace $\mathcal{P}(X = v \mid \mathrm{Pa}(X) = b)$ with $\mathcal{P}(X = v)$ in the above argument. $\qquad\square$

**Proposition 5.** *For any non-leaf node $X$ and label $b$:*

- *$\Phi_{\mathrm{Ch}(X),v}(m) = \Phi_{\mathrm{Ch}(X),v}(0) + h_{\mathrm{Ch}(X),v}(m)$ for some piecewise linear $h_{\mathrm{Ch}(X),v}(m)$.*

- *$\Phi_{\mathrm{Ch}(X),v}(m) = m$ if and only if $m \geq \max_{Y \in \mathrm{Ch}(X)} g(Y, b)$.*

*Proof.* For the first item, recall that we showed that $\int_m^{\frac{\bar{r}}{1-\beta}} \prod_{Y \in \mathrm{Ch}(X)} \frac{\partial \phi_{Y,b}(k)}{\partial k} \, dk$ is a piecewise linear function of $m$ in the proof in Lemma 4. Defining this function as $h_{\mathrm{Ch}(X),b}(m)$, Eq. (2) yields $\Phi_{\mathbf{S},b}(m) = \frac{\bar{r}}{1-\beta} + h_{\mathrm{Ch}(X),b}(m)$. So, $\Phi_{\mathbf{S},b}(0) = \frac{\bar{r}}{1-\beta} + h_{\mathrm{Ch}(X),b}(0)$, which is a constant since $h_{\mathrm{Ch}(X),b}(0)$ with respect to $m$.

For the second item, we recall the following fact from [Whi80] that $\frac{\partial \phi_{X,b}(k)}{\partial k} = \mathbb{E}[\beta^T]$, where the expectation is taken under the optimal policy when given a fallback option $m$ and $T$ is the optimal stopping time for node $X$. Since $\beta \in (0, 1)$, we see that $\mathbb{E}[\beta^T] \leq 1$ with equality if and only if when $T = 0$, which happens only when $m \geq g(X, b)$. So, the product in Eq. (2) $\prod_{Y \in \mathrm{Ch}(X)} \frac{\partial \phi_{Y,b}(k)}{\partial k} \, dk \leq 1$ with equality if and only if $k \geq \max_{Y \in Ch(X)} g(Y, b)$. Meanwhile, when the product equals to 1, we get $\Phi_{\mathrm{Ch}(X),b}(m) = \frac{\bar{r}}{1-\beta} - \int_m^{\frac{\bar{r}}{1-\beta}} 1 \, dk = m$. Therefore, $\Phi_{\mathrm{Ch}(X),b}(m) = m$ if and only if $m \geq \max_{Y \in \mathrm{Ch}(X)} g(Y, b)$. $\qquad\square$

For convenience of proving Theorem 6, let us recall Eq. (1) and Eq. (2) from the main text:

To define the Gittins index, let us first define two recursive functions $\phi$ and $\Phi$, as per [KO03]. For any non-root node $X \in \mathbf{X}$, label $b \in \mathbf{\Omega}$, and value $0 \le m \le \frac{\bar{r}}{1-\beta}$,

$$\phi_{X,b}(m) = \max\left\{m, \sum_{v\in\mathbf{\Omega}} \mathcal{P}(X=v \mid \mathrm{Pa}(X)=b) \cdot \left[r(X,v)+\beta\cdot\Phi_{\mathrm{Ch}(X),v}(m)\right]\right\}$$
$$(1)$$

If $X$ is the root, we define $\phi_{X,\emptyset}(m) = \max\left\{m, \sum_{v\in\mathbf{\Omega}} \mathcal{P}(X=v) \cdot \left[r(X,v) + \beta \cdot \Phi_{\mathrm{Ch}(X),v}(m)\right]\right\}$. For any subset of nodes $\mathbf{S} \in \mathbf{X}$, label $b \in \mathbf{\Omega}$, and value $0 \le m \le \frac{\bar{r}}{1-\beta}$,

$$\Phi_{\mathbf{S},b}(m) = \begin{cases} \frac{\bar{r}}{1-\beta} - \int_m^{\frac{\bar{r}}{1-\beta}} \prod_{Y\in\mathbf{S}} \frac{\partial\phi_{Y,b}(k)}{\partial k} \, dk & \text{if } \mathbf{S} \neq \emptyset \\ m & \text{if } \mathbf{S} = \emptyset \end{cases} \qquad (2)$$

We will only invoke Eq. (2) with $\mathbf{S} = \mathrm{Ch}(X)$ for some node $X$.

**Theorem 6.** *Given graph $\mathcal{G} = (\mathbf{X}, \mathbf{E})$ and oracle access[15] to joint distribution $\mathcal{P}$, the Gittins indices can be computed in $O(n^2 \cdot |\mathbf{\Omega}|^2)$ time while using $O(n \cdot |\mathbf{\Omega}|^2)$ oracle calls to $\mathcal{P}$ via Algorithm 1. The space complexity is $O(n^2 \cdot |\mathbf{\Omega}|)$ space for storing $O(n \cdot |\mathbf{\Omega}|)$ intermediate piecewise linear functions.*

*Proof.* Without loss of generality, we may assume that there is only one rooted tree since the computation for each connected component is independent For instance, if there are $k$ components and the $i$-th component has $n_i$ nodes, then the overall complexity is $\mathcal{O}(\sum_{i=1}^{k} n_i^2 \cdot |\mathbf{\Omega}|^2) \subseteq \mathcal{O}(n^2 \cdot |\mathbf{\Omega}|^2)$.

Throughout this proof, we assume that any conditional probability value can be obtained in constant time via oracle access to $\mathcal{P}$. Now, recalling the definitions of Eq. (1) and Eq. (2), and Lemma 4, we know that the computation of $\phi$ and $\Phi$ involve manipulating piecewise linear functions. The proof outline is as follows: we first use induction to argue that for any node $X$, the set of functions $\{\phi_{X,b}\}_{b\in\mathbf{\Omega}}$ can be computed using $O(|\mathbf{\Omega}|^2)$ oracle calls to $\mathcal{P}$ and $O(|\mathbf{\Omega}| \cdot \max\{1, |\mathrm{Ch}(X)|\})$ operations on piecewise linear functions. Then, we argue that the maximum time to perform any piecewise linear function operation in Algorithm 1 is upper bounded by $O(n \cdot |\mathbf{\Omega}|)$. Our claim follows by summing over all nodes $X$ and using the upper bound cost of operating on piecewise linear function.

---

By inducting on the nodes from the leaf towards the root, we first show that the set of functions $\{\phi_{X,b}\}_{b\in\mathbf{\Omega}}$ can be computed using $O(|\mathbf{\Omega}|^2)$ oracle calls to $\mathcal{P}$ and $O(|\mathbf{\Omega}| \cdot \max\{1, |\mathrm{Ch}(X)|\})$ operations on piecewise linear functions, as long as we store intermediate functions $\{\phi_{Y,b}\}_{Y\in\mathrm{Ch}(X),b\in\mathbf{\Omega}}$ along the way. As a reminder, in this part of the proof, we are abstracting away the computation cost for manipulating pieces of piecewise functions and focus on counting the number of operations on piecewise functions; we will later upper bound the computational cost for each of these operations.

**Base case ($X$ is a leaf):** Recall from the proof of Lemma 4 that the function $\phi_{X,b}$ is defined as

$$\phi_{X,b}(m) = \max\left\{m, \beta m + \sum_{v\in\mathbf{\Omega}} \mathcal{P}(X=v \mid \mathrm{Pa}(X)=b) \cdot r(X,v)\right\}$$

for any label value $b \in \mathbf{\Omega}$. Since $\sum_{v\in\mathbf{\Omega}} \mathcal{P}(X=v \mid \mathrm{Pa}(X)=b)\cdot r(X,v)$ can be computed in $O(|\mathbf{\Omega}|)$ oracle calls to $\mathcal{P}$, the function $\phi_{X,b}$ can be computed with $O(1)$ further operations on piecewise linear functions. So, the set of functions $\{\phi_{X,b}\}_{b\in\mathbf{\Omega}}$ can be computed in $O(|\mathbf{\Omega}|^2)$ oracle calls to $\mathcal{P}$ and $O(|\mathbf{\Omega}|)$ operations on piecewise linear functions.

**Inductive case ($X$ is not a leaf, i.e. $\mathrm{Ch}(X) \neq \emptyset$):** Suppose all children nodes $Y \in \mathrm{Ch}(X)$ of $X$ have computed and stored their piecewise linear functions $\phi_{Y,v}$ for all possible values $v \in \mathbf{\Omega}$.

Fix an arbitrary label $b \in \mathbf{\Omega}$. To compute $\Phi_{X,b}$ we need $O(|\mathrm{Ch}(X)|)$ operations on piecewise linear functions to differentiate each $\phi_{Y,b}$ function, and then multiply them together. Integrating

---

[15]The focus is on the recursive cost of computing Gittins indices and the oracle access assumption is meant to abstract away the computational cost of computing quantities like $\mathcal{P}(\cdot \mid \cdot)$. As our setting assumes that $\mathcal{P}$ is a pairwise MRF defined over a tree-structured graph, such inference is tractable via the junction tree algorithm.

this resultant function and subtracting it from the constant $\frac{\bar{r}}{1-\beta}$ function requires only an additional $O(1)$ operations on piecewise linear functions. Thus, computing all functions $\{\Phi_{\mathrm{Ch}(X),b}\}_{b\in\mathbf{\Omega}}$ costs $O(|\mathrm{Ch}(X)|\cdot|\mathbf{\Omega}|)$ operations on piecewise linear functions.

Fix an arbitrary label $b\in\mathbf{\Omega}$. To compute $\phi_{X,b}$, we need to manipulate the set of functions $\{\Phi_{\mathrm{Ch}(X),v}\}_{v\in\mathbf{\Omega}}$. More precisely, we use $O(1)$ operations to scale each $\Phi_{\mathrm{Ch}(X),v}$ function by a constant $\beta$, add the constant $r(X,v)$ function to each of them, and multiply again by the constant value $\mathcal{P}(X=v\mid\mathrm{Pa}(X)=b)$. Note that each $\mathcal{P}(X=v\mid\mathrm{Pa}(X)=b)$ can be obtain in a single call to the $\mathcal{P}$ oracle, i.e. a total of $O(|\mathbf{\Omega}|)$ calls. A further $O(|\mathbf{\Omega}|)$ operations on piecewise linear functions suffice to sum these manipulated functions up and take the maximum against the linear $m$ function. So, the entire set of functions $\{\phi_{X,b}\}_{b\in\mathbf{\Omega}}$ can be computed in $O(|\mathbf{\Omega}|^2)$ oracle calls to $\mathcal{P}$ and $O(|\mathbf{\Omega}|\cdot|\mathrm{Ch}(X)|)$ operations on piecewise linear functions.

**Intermediate conclusion:** By the inductive argument above, we showed the set of functions $\{\phi_{X,b}\}_{b\in\mathbf{\Omega}}$ can be computed using $O(|\mathbf{\Omega}|^2)$ oracle calls to $\mathcal{P}$ and $O(\max\{1,|\mathrm{Ch}(X)|\}\cdot|\mathbf{\Omega}|)$ operations on piecewise linear functions for any node $X$. Summing across all nodes, this incurs $O(n\cdot|\mathbf{\Omega}|^2)$ oracle calls to $\mathcal{P}$ and $O(|\mathbf{\Omega}|\cdot\sum_{X\in\mathbf{X}}\max\{1,|\mathrm{Ch}(X)|\})$ operations on piecewise linear functions.

---

It remains to argue that any operation on piecewise linear function in Algorithm 1 requires $O(n\cdot|\mathbf{\Omega}|)$ time. First, observe that the computation time for any piecewise function operation depends on the number of pieces. Let us denote the number of pieces in a piecewise function $f$ by $\#\mathrm{pieces}(f)$. Any addition or multiplication operation of two piecewise linear functions $f$ and $g$ creates a new piecewise linear function $h$ such that $\#\mathrm{pieces}(h)\leq\#\mathrm{pieces}(f)+\#\mathrm{pieces}(g)$. Differentiating and integrating a piecewise linear function do not change the number of pieces. Finally, taking the max of a piecewise linear against a linear function can at most increase the number of pieces by 1. From Lemma 8, we know that the *maximum* number of pieces in any piecewise linear operation involving in the computation of Algorithm 1 is at most $O(n\cdot|\mathbf{\Omega}|)$.

Putting together everything, we see that Algorithm 1 incurs $O(n\cdot|\mathbf{\Omega}|^2)$ oracle calls to $\mathcal{P}$ and $O(|\mathbf{\Omega}|\cdot\sum_{X\in\mathbf{X}}\max\{1,|\mathrm{Ch}(X)|\})$ operations on piecewise linear functions. Since each operation on piecewise linear functions costs at most $O(n\cdot|\mathbf{\Omega}|)$, Algorithm 1 runs in $O(n^2\cdot|\mathbf{\Omega}|^2)$ time while using $O(n\cdot|\mathbf{\Omega}|^2)$ oracle calls to $\mathcal{P}$. $\qquad\square$

## F.2 Parameter fitting

For convenience of proving Lemma 9, let us recall Eq. (11) from Appendix C.2:

$$\widetilde{\mathcal{P}}_{\theta_1,\theta_2}(\mathbf{x})=\prod_{i=1}^{n}\mathcal{P}_{\theta_1,\theta_2}(X_i=x_i\mid\mathbf{x}_{-i})=\prod_{i=1}^{n}\frac{\mathcal{P}_{\theta_1,\theta_2}(\mathbf{x})}{\mathcal{P}_{\theta_1,\theta_2}(X_i=0,\mathbf{x}_{-i})+\mathcal{P}_{\theta_1,\theta_2}(X_i=1,\mathbf{x}_{-i})}$$

where

$$\mathcal{P}_{\theta_1,\theta_2}(\mathbf{X}=\mathbf{x})=\frac{1}{z(\theta_1,\theta_2)}\exp\left(\sum_{i=1}^{n}\theta_1^\top f_1(x_i,\mathbf{c}_i)+\sum_{\{i,j\}\in\mathbf{E}}\theta_2^\top f_2(x_i,x_j,\mathbf{c}_i,\mathbf{c}_j)\right)$$

and

$$\phi_i(x_i)=\exp\left(\theta_1^\top f_1(x_i,\mathbf{c}^{(i)})\right)\quad\text{and}\quad\phi_{i,j}(x_i,x_j)=\exp\left(\theta_2^\top f_2(x_i,x_j,\mathbf{c}^{(i)},\mathbf{c}^{(j)})\right)$$

**Lemma 9.** *Let $\mathcal{G}=(\mathbf{X},\mathbf{E})$ be a graph over $n$ nodes, where each node $X_i\in\mathbf{X}$ has binary label $x_i\in\{0,1\}$, covariates $\mathbf{c}_i\in\mathbb{R}^d$, and neighborhood $\mathbf{N}(X_i)$. Define feature maps $f_1(x_i,\mathbf{c}_i)$ and $f_2(x_i,x_j,\mathbf{c}_i,\mathbf{c}_j)$ as per Eq. (8) and Eq. (9) respectively, shared parameters $\theta_1\in\mathbb{R}^{2+2d}$ and $\theta_2\in\mathbb{R}^{4+5d}$, and the joint probability as per Eq. (10). Then, the log-pseudolikelihood gradients are:*

$$\frac{\partial\log\widetilde{\mathcal{P}}_{\theta_1,\theta_2}(\mathbf{x})}{\partial\theta_1}=\sum_{i=1}^{n}\alpha_i\cdot\left(f_1(1,\mathbf{c}^{(i)})-f_1(0,\mathbf{c}^{(i)})\right)$$

$$\frac{\partial\log\widetilde{\mathcal{P}}_{\theta_1,\theta_2}(\mathbf{x})}{\partial\theta_2}=\sum_{i=1}^{n}\alpha_i\cdot\sum_{X_j\in\mathbf{N}(X_i)}\left(f_2(1,x_j,\mathbf{c}^{(i)},\mathbf{c}^{(j)})-f_2(0,x_j,\mathbf{c}^{(i)},\mathbf{c}^{(j)})\right)$$

*where the common coefficient $\alpha_i = x_i - \mathcal{P}_{\theta_1,\theta_2}(X_i = 1 \mid \mathbf{x}_{-i})$ can be computed efficiently* without *computing $z(\theta_1, \theta_2)$ for all $i \in [n]$.*

*Proof.* Let us define the terms $A$ and $B^{y,b}$ for $y \in [n]$ and $b \in \{0, 1\}$:

$$A = \log\left( z(\theta_1, \theta_2) \cdot \mathcal{P}_{\theta_1,\theta_2}(\mathbf{x}) \right) \qquad = \sum_{i=1}^{n} \theta_1^\top f_1(x_i, \mathbf{c}_i) + \sum_{\{i,j\}\in\mathbf{E}} \theta_2^\top f_2(x_i, x_j, \mathbf{c}^{(i)}, \mathbf{c}^{(j)})$$

$$B^{y,b} = \log\left( z(\theta_1, \theta_2) \cdot \mathcal{P}_{\theta_1,\theta_2}(X_y = b, \mathbf{x}_{-y}) \right) \quad = \theta_1^\top f_1(b, \mathbf{c}^{(y)}) + \sum_{\substack{\{i,j\}\in\mathbf{E} \\ y=i}} \theta_2^\top f_2(b, x_j, \mathbf{c}^{(y)}, \mathbf{c}^{(j)})$$

$$+ \sum_{\substack{i=1 \\ i\neq y}}^{n} \theta_1^\top f_1(x_i, \mathbf{c}^{(i)}) + \sum_{\substack{\{i,j\}\in\mathbf{E} \\ y\notin\{i,j\}}} \theta_2^\top f_2(x_i, x_j, \mathbf{c}^{(i)}, \mathbf{c}^{(j)})$$

Observe that

$$\frac{\exp(B_y^1)}{\exp(B_y^0) + \exp(B_y^1)} = \mathcal{P}_{\theta_1,\theta_2}(\mathbf{X}_y = 1 \mid \mathbf{x}_{-y}) \tag{12}$$

Let us consider their partial differentiation with respect to $\theta_1$ and $\theta_2$. We will use these later.

$$\frac{\partial A}{\partial \theta_1} = \sum_{i=1}^{n} \frac{\theta_1^\top f_1(x_i, \mathbf{c}^{(i)})}{\partial \theta_1} \tag{13}$$

$$= \sum_{i=1}^{n} f_1(x_i, \mathbf{c}^{(i)})$$

$$\frac{\partial A}{\partial \theta_2} = \sum_{\{i,j\}\in\mathbf{E}} \frac{\partial \theta_2^\top f_2(x_i, x_j, \mathbf{c}^{(i)}, \mathbf{c}^{(j)})}{\partial \theta_2} \tag{14}$$

$$= \sum_{\{i,j\}\in\mathbf{E}} f_2(x_i, x_j, \mathbf{c}^{(i)}, \mathbf{c}^{(j)})$$

$$\frac{\partial B_y^b}{\partial \theta_1} = \frac{\partial \theta_1^\top f_1(b, \mathbf{c}_y)}{\partial \theta_1} + \sum_{\substack{i=1 \\ i\neq y}}^{n} \frac{\partial \theta_1^\top f_1(x_i, \mathbf{c}^{(i)})}{\partial \theta_1} \tag{15}$$

$$= f_1(b, \mathbf{c}_y) + \sum_{\substack{i=1 \\ i\neq y}}^{n} f_1(x_i, \mathbf{c}^{(i)})$$

$$\frac{\partial B_y^b}{\partial \theta_2} = \sum_{\substack{\{i,j\}\in\mathbf{E} \\ y=i}} \frac{\partial \theta_2^\top f_2(b, x_j, \mathbf{c}^{((y)}, \mathbf{c}^{(j)})}{\partial \theta_2} + \sum_{\substack{\{i,j\}\in\mathbf{E} \\ y\notin\{i,j\}}} \frac{\partial \theta_2^\top f_2(x_i, x_j, \mathbf{c}^{(i)}, \mathbf{c}^{(j)})}{\partial \theta_2} \tag{16}$$

$$= \sum_{\substack{\{i,j\}\in\mathbf{E} \\ y=i}} f_2(b, x_j, \mathbf{c}^{(y)}, \mathbf{c}^{(j)}) + \sum_{\substack{\{i,j\}\in\mathbf{E} \\ y\notin\{i,j\}}} f_2(x_i, x_j, \mathbf{c}^{(i)}, \mathbf{c}^{(j)})$$

Now, re-expressing $\widetilde{\mathcal{P}}_{\theta_1,\theta_2}(\mathbf{x})$ in terms of $A$ and $B_y^b$, we have

$$\widetilde{\mathcal{P}}_{\theta_1,\theta_2}(\mathbf{x}) = \prod_{i=1}^{n} \frac{\mathcal{P}_{\theta_1,\theta_2}(\mathbf{x})}{\mathcal{P}_{\theta_1,\theta_2}(X_i = 0, \mathbf{x}_{-i}) + \mathcal{P}_{\theta_1,\theta_2}(X_i = 1, \mathbf{x}_{-i})} = \prod_{y=1}^{n} \frac{\exp(A)}{\exp(B_y^0) + \exp(B_y^1)}$$

Fix an index $y \in [n]$ and define $W_y = \frac{\exp(A)}{\exp(B_y^0)+\exp(B_y^1)}$ so that $\widetilde{\mathcal{P}}_{\theta_1,\theta_2}(\mathbf{x}) = \prod_{y=1}^{n} W_y$, i.e. $\log \widetilde{\mathcal{P}}_{\theta_1,\theta_2}(\mathbf{x}) = \sum_{y=1}^{n} \log W_y$.

Let us differentiate with respect to $\theta_1$. For any $y \in [n]$, we see that

$$\frac{\partial \log W_y}{\partial \theta_1}$$

$$= \frac{\partial A}{\partial \theta_1} - \frac{\partial \log(\exp(B_y^0) + \exp(B_y^1))}{\partial \theta_1}$$

$$= \frac{\partial A}{\partial \theta_1} - \frac{1}{\exp(B_y^0) + \exp(B_y^1)} \left( \frac{\partial \exp(B_y^0)}{\partial \theta_1} + \frac{\partial \exp(B_y^1)}{\partial \theta_1} \right)$$

$$= \frac{\partial A}{\partial \theta_1} - \frac{\exp(B_y^0)}{\exp(B_y^0) + \exp(B_y^1)} \frac{\partial B_y^0}{\partial \theta_1} - \frac{\exp(B_y^1)}{\exp(B_y^0) + \exp(B_y^1)} \frac{\partial B_y^1}{\partial \theta_1}$$

$$= \frac{\partial A}{\partial \theta_1} - (1 - \mathcal{P}_{\theta_1, \theta_2}(\mathbf{X}_y = 1 \mid \mathbf{x}_{-y})) \frac{\partial B_y^0}{\partial \theta_1} - \mathcal{P}_{\theta_1, \theta_2}(\mathbf{X}_y = 1 \mid \mathbf{x}_{-y})) \frac{\partial B_y^1}{\partial \theta_1} \qquad \text{(By Eq. (12))}$$

$$= \sum_{i=1}^{n} f_1(x_i, \mathbf{c}_i) - (1 - \mathcal{P}_{\theta_1, \theta_2}(\mathbf{X}_y = 1 \mid \mathbf{x}_{-y})) \left( f_1(0, \mathbf{c}_y) + \sum_{\substack{i=1 \\ i \neq y}}^{n} f_1(x_i, \mathbf{c}_i) \right)$$

$$- \mathcal{P}_{\theta_1, \theta_2}(\mathbf{X}_y = 1 \mid \mathbf{x}_{-y}) \left( f_1(1, \mathbf{c}_y) + \sum_{\substack{i=1 \\ i \neq y}}^{n} f_1(x_i, \mathbf{c}_i) \right)$$

$$\text{(By Eq. (13) and Eq. (15))}$$

$$= f_1(x_y, \mathbf{c}_y) - (1 - \mathcal{P}_{\theta_1, \theta_2}(\mathbf{X}_y = 1 \mid \mathbf{x}_{-y})) \cdot f_1(0, \mathbf{c}_y) - \mathcal{P}_{\theta_1, \theta_2}(\mathbf{X}_y = 1 \mid \mathbf{x}_{-y}) \cdot f_1(1, \mathbf{c}_y)$$

$$= (x_y - \mathcal{P}_{\theta_1, \theta_2}(\mathbf{X}_y = 1 \mid \mathbf{x}_{-y})) \cdot (f_1(1, \mathbf{c}_y) - f_1(0, \mathbf{c}_y)) \qquad \text{(Since } x_y \in \{0, 1\})$$

Summing over all $y \in [n]$, we get

$$\frac{\partial \log \widetilde{\mathcal{P}}_{\theta_1, \theta_2}(\mathbf{x})}{\partial \theta_1} = \sum_{y=1}^{n} \frac{\partial \log W_y}{\partial \theta_1} = \sum_{y=1}^{n} (x_y - \mathcal{P}_{\theta_1, \theta_2}(\mathbf{X}_y = 1 \mid \mathbf{x}_{-y})) \cdot (f_1(1, \mathbf{c}_y) - f_1(0, \mathbf{c}_y))$$

yielding the first statement as desired, where $\alpha_y = (x_y - \mathcal{P}_{\theta_1, \theta_2}(\mathbf{X}_y = 1 \mid \mathbf{x}_{-y}))$.

For the second statement, we do the same analysis but use Eq. (14) and Eq. (16) instead of Eq. (13) and Eq. (15). Note that the second summation $\sum_{\substack{\{i,j\} \in \mathbf{E} \\ y \notin \{i,j\}}}$ gets cancelled out as in the above analysis while the first summation $\sum_{\substack{\{i,j\} \in \mathbf{E} \\ y=i}}$ is exactly corresponds to $\sum_{X_j \in N(X_y)}$. $\qquad \square$

### F.3 Robustness to model misspecification

We analyze how errors in the learned graphical model impact the quality of the resulting policy in an AFEG instance. Specifically, we derive a bound on the discrepancy between the optimal state–action value functions of the *learned* and the *true* Markov decision processes (MDPs), capturing how inaccuracies in the estimated joint distribution $\hat{\mathcal{P}}$ affect decision quality.

Recall that an AFEG instance is defined by a triple $(\mathcal{G}, \mathcal{P}, \beta)$, where $\mathcal{G} = (\mathbf{X}, \mathbf{E})$ is a graph, $\mathcal{P}$ is a joint distribution over binary node labels Markov with respect to $\mathcal{G}$, and $\beta \in (0, 1)$ is a discount factor. At each time step $t$, the state $\mathcal{S}_t$ records the current frontier and the set of revealed labels. A policy $\pi$ selects a node from the frontier and receives a label-dependent reward.

We consider two MDPs defined over the same state space $\mathcal{S}$, action mapping $\mathcal{A} : \mathcal{S} \to 2^{\mathbf{X}}$ that restricts valid actions to untesed frontier nodes, and discount factor $\beta$, but differing in the distribution used to infer infection status:

- The **learned MDP** $M = (\mathcal{S}, \mathcal{A}, \hat{P}, \hat{R}, \beta)$ is defined using an estimated model $\hat{\mathcal{P}}$ obtained via pseudo-likelihood. The expected reward for testing node $a \in \mathcal{A}(s)$ is

$$\hat{R}(s, a) = \hat{\mathcal{P}}(X_a = 1 \mid s),$$

    and the transition kernel $\hat{P}$ uses this posterior to sample a binary outcome.

- The **true MDP** $M' = (\mathcal{S}, \mathcal{A}, P, R, \beta)$ is induced by the ground-truth distribution $\mathcal{P}$, with

$$R(s, a) = \mathcal{P}(X_a = 1 \mid s),$$

and transitions $P$ differing from $\hat{P}$ only in the Bernoulli parameter used for $X_a$.

We define the maximum reward and transition discrepancies:

$$\varepsilon_R = \max_{s \in \mathcal{S}, \, a \in \mathcal{A}(s)} \left| \hat{R}(s, a) - R(s, a) \right|, \qquad \varepsilon_P = \max_{s \in \mathcal{S}, \, a \in \mathcal{A}(s), \, s' \in \mathcal{S}} \left| \hat{P}(s' \mid s, a) - P(s' \mid s, a) \right|.$$

Since rewards are probabilities, we set $R_{\max} = 1$.

We are interested in bounding the worst-case deviation in optimal $Q$-values:

$$\|Q_M^* - Q_{M'}^*\|_\infty = \max_{s \in \mathcal{S}, \, a \in \mathcal{A}(s)} |Q_M^*(s, a) - Q_{M'}^*(s, a)|.$$

This is because the $Q^*$-function encodes the expected total discounted reward starting from state $s$ and taking action $a$, and thus directly characterizes the long-term value of testing each node. Therefore, a small bound on $\|Q_M^* - Q_{M'}^*\|_\infty$ ensures that the policy derived from the learned model will perform nearly as well as the optimal policy under the true model, in terms of accumulated reward. The following bound shows that the suboptimality of the learned policy is controlled by the maximum error in posterior infection probabilities and transition dynamics, with greater sensitivity as $\beta \to 1$.

**Lemma 10.** *Let $M$ and $M'$ be defined as above, and let $\varepsilon_R, \varepsilon_P$ denote the maximal reward and transition discrepancies. Then:*

$$\|Q_M^* - Q_{M'}^*\|_\infty \leq \frac{\varepsilon_R}{1 - \beta} + \frac{\beta R_{\max}}{(1 - \beta)^2} \varepsilon_P.$$

*In particular, since $R_{\max} = 1$,*

$$\|Q_M^* - Q_{M'}^*\|_\infty \leq \frac{1}{1 - \beta} \left( \varepsilon_R + \frac{\beta}{1 - \beta} \varepsilon_P \right).$$

*Proof.* Let $T_M$ and $T_{M'}$ denote the Bellman optimality operators for MDPs $M$ and $M'$ respectively. For any function $Q : \mathcal{S} \times \mathcal{A} \to \mathbb{R}$, define:

$$(T_M Q)(s, a) = R(s, a) + \beta \sum_{s' \in \mathcal{S}} P(s' \mid s, a) \max_{a'} Q(s', a')$$

The same holds for the operator $T_{M'}$. We also know that the operators $T_M$ and $T_{M'}$ are both $\beta$-contractions under the supremum norm:

$$\|T_M Q - T_M Q'\|_\infty \leq \beta \|Q - Q'\|_\infty \quad \text{and} \quad \|T_{M'} Q - T_{M'} Q'\|_\infty \leq \beta \|Q - Q'\|_\infty$$

Now, let $Q_M^*$ and $Q_{M'}^*$ be the fixed points of $T_M$ and $T_{M'}$ respectively. Then,

$$
\begin{aligned}
\|Q_M^* - Q_{M'}^*\|_\infty &= \|T_M Q_M^* - T_{M'} Q_{M'}^*\|_\infty && (Q_M^* \text{ and } Q_{M'}^* \text{ are fixed points}) \\
&\leq \|T_M Q_M^* - T_{M'} Q_M^*\|_\infty + \|T_{M'} Q_M^* - T_{M'} Q_{M'}^*\|_\infty && \text{(Triangle inequality)} \\
&\leq \|T_M Q_M^* - T_{M'} Q_M^*\|_\infty + \beta \|Q_M^* - Q_{M'}^*\|_\infty && \text{(Contraction property of } T_{M'})
\end{aligned}
$$

Rearranging, we get

$$(1 - \beta)\|Q_M^* - Q_{M'}^*\|_\infty \leq \|T_M Q_M^* - T_{M'} Q_M^*\|_\infty \tag{17}$$

As the difference $\|T_M Q_M^* - T_{M'} Q_M^*\|_\infty$ captures how much the two MDPs' rewards and transitions differ, we will analyze and bound for $(T_M Q)(s, a) - (T_{M'} Q)(s, a)$ for some fixed $Q$:

$$
\begin{aligned}
(T_M Q)(s, a) &- (T_{M'} Q)(s, a) \\
&= \left( R(s, a) - \hat{R}(s, a) \right) + \beta \sum_{s'} \left( P(s' \mid s, a) - \hat{P}(s' \mid s, a) \right) \max_{a'} Q(s', a').
\end{aligned}
$$

Taking absolute values and using the definition of $\varepsilon_R$ and $\varepsilon_P$, we get

$$|(T_M Q)(s, a) - (T_{M'} Q)(s, a)| \leq \varepsilon_R + \beta \varepsilon_P \|Q\|_\infty.$$

Therefore,
$$\|T_M Q - T_{M'} Q\|_\infty \le \varepsilon_R + \beta \varepsilon_P \|Q\|_\infty.$$
In particular, setting $Q = Q_{M'}^*$ and using the bound $\|Q_{M'}^*\|_\infty \le R_{\max}/(1 - \beta)$, we see that

$$\|T_M Q_{M'}^* - T_{M'} Q_{M'}^*\|_\infty \le \varepsilon_R + \frac{\beta R_{\max}}{1 - \beta} \varepsilon_P$$

Plugging into Eq. (17), we get

$$(1 - \beta)\|Q_M^* - Q_{M'}^*\|_\infty \le \varepsilon_R + \frac{\beta R_{\max}}{1 - \beta} \varepsilon_P$$

Thus,

$$\|Q_M^* - Q_{M'}^*\|_\infty \le \frac{\varepsilon_R}{1 - \beta} + \frac{\beta R_{\max}}{(1 - \beta)^2} \varepsilon_P$$

as desired. $\qquad\square$

