# OpenReview forum: "Adaptive Frontier Exploration on Graphs with Applications to Network-Based Disease Testing"
_NeurIPS.cc/2025/Conference — NeurIPS 2025 poster_

### Official Review · Reviewer_ALF3 · 2025-06-11

**Clarity:** 3
**Significance:** 3
**Originality:** 3
**Rating:** 5
**Confidence:** 3

**Summary:**

The authors propose the adaptive frontier graph exploration problem, whereby one wants to explode the nodes of a graph in which each node has a label, each label a reward. The goal is to maximize total expected (discounted) reward. The formulation essentially combines a Markov decision process with dynamic graph exploration.

They develop a Gittins-index approach, proving that their algorithm is optimal in the case of trees, and showing experimentally that it performs well in more complicated, real-world graphs based on disease networks. They also prove several new results concerning the computational tractability of the Gittins-index in their setting.

**Questions:**

1. Does the agent know the structure of the graph a priori? There are flavors of the graph exploration problem where the graph structure has to be discovered by exploring.

2. I find the formulation of $\pi^*$ a bit confusing. You seem to be using $X_\pi$ to denote both the node itself and the node’s label (otherwise how can we have $X_\pi = v$?). But the reward function treats $X_\pi$ as the node itself, not the label. (Eg  in the public health example at the bottom of page 2 you treat $X$ as an individual). I’m not sure you should necessarily change anything, but if you agree with this, it might be worth commenting on.

3. Is it realistic to assume that n is known in advance? In the public health example, it seems you might start exploring and only decide to stop based on what you find, or external factors such as budget etc. But it doesn’t seem likely that you’d commit to only testing n = 10,000 and, regardless of the outcomes, stop after that.

4. You say: “Our work addresses this gap by presenting the first efficient implementation of Gittins-based policies in  discrete branching bandits with history-dependent rewards, enabling their use in structured settings like network-based disease testing.” (Pg 4). How general are the structured settings that you’re alluding to? If very general, one wonders why you’re focused on adaptive graph exploration the first place, and didn’t write the paper about the more general setting of discrete branching bandits. A sentence or two about this would be helpful for readers like me, who aren’t extremely familiar with this area.

5. In Figure 4, it’s interesting that greedy outperforms DQN in ~half the experiments. This seems counterintuitive? Also, if you increase the number of extra edges, does the Gittins performance ever fall below greedy and/or DQN, or do they simply converge?

**Minor comments**:
- Researchers typo in line 200.

**Ethical Concerns:**

["NO or VERY MINOR ethics concerns only"]

**Final Justification:**

Maintaining my original score.

**Limitations:**

Yes

**Quality:**

4

**Strengths And Weaknesses:**

# Strengths
- Good results, both on synthetic data and real-world data
- Well motivated problem setting. The authors did a good job (modulo one complaint below) of tying the formalism back to a real world setting.
- New results on the computational tractability of the Gittins-index.
- Well written

# Weaknesses
- The mathematical details are a bit hard to follow for a non-expert in the area. The Gittins formulation in particular is not intuitive, nor do the authors spend that much time providing intuition. But I understand that space constraints make this challenging.
- I find the generality of the result a bit hard to parse (see below). It seems that they solve a problem that may be more general than adaptive graph exploration. (This is a minor weakness; could even be a strength if addressed.)
- I'd like to see the experimental results stretched a bit more - when does their algorithm start being worse than its competitors? Surely there are some scenarios in which it is not the best.

---

> ### Author Rebuttal · Authors · 2025-07-31
>
> Thank you for your time and effort for reviewing our work, and for your thoughtful feedback. We appreciate the opportunity to address your concerns in this rebuttal and we will add a version of the discussion below in our revision.
>
> # Gittins index computation example
>
> We agree that the mathematical details of Gittins index computation can be difficult to follow, especially without prior familiarity. To provide better intuition, we will add the following worked example (with illustration) to the appendix of our revision. (Note: We're sorry that the "align" and "cases" are not rendering well on OpenReview...)
>
> > Consider a graph G on 7 nodes with edges $\mathbf{E} = \bigg\\{ \\{X_0, X_1\\}, \\{X_0, X_2\\}, \\{X_0, X_3\\}, \\{X_2, X_4\\}, \\{X_2, X_5\\}, \\{X_3, X_6\\} \bigg\\}$. Let $X_0$ be the root node in this graph.
>
> > Consider the following simple joint binary distribution $\mathcal{P}$ which assigns higher probabilities to realizations where adjacent nodes have the same status:
> $$
> \mathcal{P}(x)
> = \frac{1}{Z} \prod_{\{X_i, X_j\}} \exp \left( 1_{X_i = X_j} \right)
> $$
> where $1_{X_i = X_j}$ is the indicator whether the statuses of adjacent nodes $X_i$ and $X_j$ agree. One can verify that we have $\mathcal{P}(u = 1 \mid v = 0) = \frac{1}{1+e}$ and $\mathcal{P}(u = 1 \mid v = 1) = \frac{e}{1+e}$ for any non-root node $u$ with parent $v$.
>
> > Now, suppose the discount factor $\beta = 0.9$ and we have a reward equal to the binary label for each node. Then, the largest reward $\overline{r} = 1$ and $0 \leq m \leq \frac{\overline{r}}{1-\beta} = 10$.
>
> > Our Gittins computation begins from the leaves. For instance, let us consider the leaf node $X_4$. When parent node $X_2 = 0$, one can verify that
> $$
> \begin{align}
> \phi_{X_4, 0}(m)
> &= \max \left\\{ m, \mathcal{P}(X_4 = 0 \mid X_2 = 0) \cdot \left[ 0 + \beta \cdot \Phi_{\emptyset, 0}(m) \right] + \mathcal{P}(X_4 = 1 \mid X_2 = 0) \cdot \left[ 1 + \beta \cdot \Phi_{\emptyset, 0}(m) \right] \right\\}\\
> &= \max \left\\{ m, \left( 1 - \frac{1}{1+e} \right) \cdot \beta \cdot m + \frac{1}{1+e} \cdot \left[ 1 + \beta \cdot m \right] \right\\}\\
> &= \max \left\\{ m, \beta m + \frac{1}{1+e} \right\\}
> \end{align}
> $$
> That is, $\phi_{X_4, 0}(m)$ is the following piecewise linear function:
> $$
> \phi_{X_4, 0}(m) =
> \begin{cases}
> \beta m + \frac{1}{1+e} & \text{if $m \leq \frac{10}{1+e}$}\\
> m & \text{if $m \geq \frac{10}{1+e}$}
> \end{cases}
> $$
>
> > Meanwhile, when parent node $X_2 = 0$, one can verify that
> $$
> \begin{align}
> \phi_{X_4, 1}(m)
> &= \max \left\\{ m, \mathcal{P}(X_4 = 0 \mid X_2 = 1) \cdot \left[ 0 + \beta \cdot \Phi_{\emptyset, 0}(m) \right] + \mathcal{P}(X_4 = 1 \mid X_2 = 1) \cdot \left[ 1 + \beta \cdot \Phi_{\emptyset, 0}(m) \right] \right\\}\\
> &= \max \left\\{ m, \left( 1 - \frac{e}{1+e} \right) \cdot \beta \cdot m + \frac{e}{1+e} \cdot \left[ 1 + \beta \cdot m \right] \right\\}\\
> &= \max \left\\{ m, \beta m + \frac{e}{1+e} \right\\}
> \end{align}
> $$
> That is, $\phi_{X_4, 1}(m)$ is the following piecewise linear function:
> $$
> \phi_{X_4, 1}(m) =
> \begin{cases}
> \beta m + \frac{e}{1+e} & \text{if $m \leq \frac{10e}{1+e}$}\\
> m & \text{if $m \geq \frac{10e}{1+e}$}
> \end{cases}
> $$
>
> > Due to symmetry in $\mathcal{P}$, all the leaf nodes have exactly the same $\phi_{X_{leaf}, b}$ functions for $b \in \{0,1\}$.
>
> > Now, let us consider the computation of the function $\Phi_{\{X_4, X_5\}, 0}$, which is required for the computation of $\phi_{X_2, \cdot}$. From Equation (2), we know that this involves the product of function derivatives $\frac{\partial \phi_{X_4, 0}(k)}{\partial k} \cdot \frac{\partial \phi_{X_5, 0}(k)}{\partial k}$. From above, one can check that this evaluates to the following piecewise *constant* function:
> $$
> \begin{cases}
> \beta^2 & \text{if $k \leq \frac{10}{1+e}$}\\
> 1 & \text{if $k \geq \frac{10}{1+e}$}
> \end{cases}
> $$
> whose integration from $0$ to $m$ yields the following piecewise linear function $h(m)$:
> $$
> h(m) =
> \begin{cases}
> \beta^2 m & \text{if $k \leq \frac{10}{1+e}$}\\
> m - \frac{10}{1 + e} (1 - \beta^2) & \text{if $k \geq \frac{10}{1+e}$}
> \end{cases}
> $$
> Using Proposition 5, we can derive that
> $$
> \begin{align}
> \Phi_{\{X_4, X_5\}, 0}(m)
> &= h(m) + 10 - h(10)\\
> &= h(m) + \frac{10}{1+e}(1 - \beta^2)\\
> &=
> \begin{cases}
> \beta^2 m + \frac{10}{1+e}(1 - \beta^2) & \text{if $k \leq \frac{10}{1+e}$}\\
> m & \text{if $k \geq \frac{10}{1+e}$}
> \end{cases}
> \end{align}
> $$
>
> > One can continue this computation up the rooted tree. Using our Gittins computation code, this produces
> $$
> \phi_{X_0, \emptyset} =
> \begin{cases}
> 0.4783 m + 2.6423 & \text{if $0 \leq m \leq 2.6894$}\\
> 0.5984 m + 2.3193 & \text{if $2.6894 \leq m \leq 3.5900$}\\
> 0.6461 m + 2.1479 & \text{if $3.5900 \leq m \leq  4.1452$}\\
> 0.7118 m + 1.8756 & \text{if $4.1452 \leq m \leq 6.5088$}\\
> m & \text{if $m \geq 6.5088$}
> \end{cases}
> $$
>
> # Stretching the experimental results
>
> Thank you for this suggestion. During the rebuttal period, we conducted additional experiments where policies are evaluated on noisy approximations $\mathcal{Q}\_{\theta}$ of the true underlying joint distribution $\mathcal{P}\_{\theta}$. These experiments shed light on when and how performance degrades due to model mismatch. Due to character limits, we refer you to the rebuttals to other reviewers for full results, and we will include this expanded analysis in our revision.
>
> # Does the agent know the graph structure?
>
> You are correct that our optimality guarantees assume full knowledge of the interaction graph $\mathcal{G}$. In real-world disease testing, $\mathcal{G}$ is often revealed incrementally: individuals are tested as they visit clinics, are interviewed by public health staff, and may refer peers via voucher programs. A practical strategy is to let this process run for a fixed period, yielding a partial observation of $\mathcal{G}$, after which AFEG can then be applied to this discovered subgraph to guide the allocation of limited testing resources. This staged approach fits naturally into our framework. Extending our method to fully online graph discovery is a compelling direction for future work.
>
> # Notation Overload
>
> We appreciate the reviewer's suggestion regarding potential ambiguity in our notation. We follow standard conventions in the graphical models literature, where variables and nodes are often used interchangeably, and the graph structure implicitly defines the factorization of the joint distribution. In this setting, it is common to use a symbol like $X$ to refer to both the node and its associated random variable, so that expressions like $X = v$ denote the event that variable $X$ takes value $v$. That said, we understand that readers from other communities may find this notation unfamiliar, and so we will include a brief remark early in the paper clarifying this convention to aid accessibility.
>
> # Knowing $n$ in advance
>
> Yes, we consider the setting where the structure of the interaction graph $\mathcal{G}$ is first obtained before testing decisions are made. Once $\mathcal{G}$ is known, the total number of nodes $n$ is fixed, and we can plan whom to engage subject to a testing budget. Exploring settings with uncertain or dynamic $\mathcal{G}$ is an interesting future direction.
>
> Importantly, our method does **not** require knowing the exact number of tests (i.e., the testing budget) in advance. Note that AFEG policies continually select an untested node given observed outcomes of tested nodes, and as a result could be seen as "anytime policies". That is, these AEFG policies can be executed sequentially and stopped at any point as resources are exhausted. In our experiments, we plot the cumulative performance of each policy as testing progresses, i.e., the total reward attained after testing 10%, 20%, …, up to 100% of the population. To assess performance under a specific budget, one can take a vertical slice of the plot at the desired percentage. For ease of comparison, we highlight the 50% mark with a dotted line in the figures in our submission, but any vertical slice yields a valid comparison. Each policy thus defines a full budget-performance tradeoff curve, and our method is not tuned for any particular testing threshold.
>
> # Generality of our result, and its relation with AFEG and branching bandits
>
> Thank you for this observation. AFEG is a structured formulation that reflects real-world network testing constraints, particularly the frontier condition. While AFEG instances are not necessarily trees, one of our key contributions (Section 3.1) is to show that Gittins index policies are optimal for AFEG when the input graph is a forest.
>
> As noted on Lines 134–140, although Gittins index policies are known to be optimal for branching bandits [KO03], no efficient implementation had been proposed previously. We believe our work is the first to provide a polynomial-time, polynomial-space implementation of Gittins indices in discrete branching bandits with history-dependent rewards, enabling their use in structured problems such as network-based disease testing.
>
> By "structured settings", we mean other domains where the problem can be reduced to a branching bandit formulation, e.g., tree-based search under uncertainty or structured sequential diagnosis. We focused on AFEG due to our public health motivating application, but agree it would be worthwhile to explore more general applications in future work.
>
> # Figure 4 and Greedy versus DQN
>
> Thank you for highlighting this. We observe that greedy outperforms DQN in some experiments, which can seem counterintuitive. One likely explanation is that DQN requires significant data to train and can overfit or underperform when training data is sparse or environments are not sufficiently varied. In contrast, greedy policies exploit strong local heuristics, which can be quite effective in certain structured networks.

---

> > ### Author Response · Authors · 2025-08-04
> >
> > Dear reviewer, I hope our response has clarified your doubts and concerns. Please let us know if there is anything else you would wish to discuss regarding our work.

---

### Official Review · Reviewer_EBjH · 2025-07-01

**Clarity:** 3
**Significance:** 2
**Originality:** 3
**Rating:** 4
**Confidence:** 2

**Summary:**

Motivated by network-based disease testing, the paper formalizes Adaptive Frontier Exploration on Graphs (AFEG) as a sequential decision problem on a probabilistic graph. When the graph is a forest, the proposed Gittins index-based policy for AFEG has a provable convergence guarantee. Experiments on both synthetic and real-world graphs show the superiority of the proposed method.

**Questions:**

1. The definitions of two potential functions $\phi_{X,b}$ and $\Phi_{S,b}$ in Eq. 1 and 2 are confusing me, although there are some attempts of explanation on L176-178. What is the difference between these potentials?
2. I am curious why the author only uses a single-step DQN as a deep RL comparison, but omits more recent graph RL or planning methods (e.g., policy-gradient, Monte-Carlo tree search). Why were they not considered?

**Ethical Concerns:**

["NO or VERY MINOR ethics concerns only"]

**Limitations:**

1. Optimality only holds for forests and relies on exact $\mathcal{P}$
2. Scalability to a very large graph is limited.

**Quality:**

3

**Strengths And Weaknesses:**

# Strengths
1. The problem of network-based disease testing is important. The proposed AFEG model correctly captures realistic locality constraints (e.g., contact-tracing).
2. The piecewise-linear analysis of the recursive value functions appears original to me and leads to the first practical algorithm for discrete branching-bandit indices.

# Weaknesses
1. Assumed access to the joint distribution $\mathcal{P}$. I think the $\mathcal{P}$ is usually learned from data in practice. How sensitive are results to model misspecification? Empirical ablations with learned vs. true $\mathcal{P}$ would strengthen the work.
2. Scalability to very large graphs. $\mathcal{O}(n^2 |\Omega|^2)$ is quadratic on $n$. Can sparsity or incremental computation be exploited to reach web-scale graphs?

---

> ### Author Rebuttal · Authors · 2025-07-31
>
> Thank you for your time and effort for reviewing our work, and for your thoughtful feedback. We appreciate the opportunity to address your concerns in this rebuttal and we will add a version of the discussion below in our revision.
>
> # Oracle access to underlying probability distribution $\\mathcal{P}$
>
> We agree that the assumption of oracle access to $\mathcal{P}$ may not always hold in practice. In real-world settings, such as modeling the spread of sexually transmitted infections (STIs), it is reasonable to approximate the distribution $\mathcal{P}$ using a pairwise Markov Random Field (MRF) over the contact network $\mathcal{G}$. As you kindly noted, we outline in Appendix B how the parameters $\theta$ of such a model can be estimated from historical data via maximum pseudo-likelihood estimation (MPLE). In addition, Appendix E.3 contains an analysis of how errors in estimating $\mathcal{P}$ affect decision quality.
>
> In the original submission, we assumed that $\mathcal{P}\_{\theta}$ (estimated via MPLE) was the ground truth. In response to your comment, we have conducted additional experiments during the rebuttal period to assess robustness when the policy only has access to a noisy version $\mathcal{Q}\_{\theta}$ of the true distribution $\mathcal{P}_{\theta}$.
>
> We perturbed the learned parameters $\theta$ with noise of magnitude $\varepsilon \in \\{0.1, 0.3, 1\\}$ to simulate distributional mismatch, while keeping the evaluation grounded in the original $\mathcal{P}\_{\theta}$. For reference, in the HIV setting, we have $\ell_{\infty}(\theta_{\text{unary}}) = 1.43$ and $\ell_{\infty}(\theta_{\text{pairwise}}) = 1.75$. Thus, $\varepsilon = 1$ constitutes a significant perturbation.
>
> Since plots and external URLs are disallowed, we summarize our results in tabular form, reporting the expected undiscounted reward over a random subset of approximately 300 nodes in the HIV network (following the format in Appendix D.3). We focus on *undiscounted* reward as it better reflects the practical goals of maximizing the number of infected individuals detected under a fixed testing budget. For clarity, we **bold** the best-performing methods for each budget level.
>
> **Expected *undiscounted* reward for $\mathcal{Q}_{\theta, 0.1}$ on HIV network**
>
> | Percentage of nodes tested | 10% | 20% | 30% | 40% | 50% | 60% | 70% | 80% | 90% | 100% |
> | :-: | :-: | :-: | :-: | :-: | :-: | :-: | :-: | :-: | :-: | :-: |
> | Random | 6.1 | 11.4 | 16.0 | 19.4 | 22.3 | 24.6 | 26.6 | 28.2 | 29.8 | **31.6** |
> | Greedy | **20.1** | 25.8 | 26.6 | 26.8 | 27.0 | 27.5 | 30.1 | 31.0 | 31.2 | **31.6** |
> | DQN | 15.7 | 22.7 | 25.4 | 26.6 | 27.0 | 27.3 | 27.9 | 28.2 | 29.9 | **31.6** |
> | Gittins | **20.1** | **28.4** | **29.4** | **30.5** | **30.9** | **31.2** | **31.3** | **31.6** | **31.6** | **31.6** |
>
> **Expected *undiscounted* reward for $\mathcal{Q}_{\theta, 0.3}$ on HIV network**
>
> | Percentage of nodes tested | 10% | 20% | 30% | 40% | 50% | 60% | 70% | 80% | 90% | 100% |
> | :-: | :-: | :-: | :-: | :-: | :-: | :-: | :-: | :-: | :-: | :-: |
> | Random | 6.1 | 11.5 | 16.0 | 19.5 | 22.3 | 24.8 | 26.6 | 28.3 | 29.8 | **31.6** |
> | Greedy | **19.2** | 25.7 | 26.4 | 26.8 | 27.0 | 27.6 | 30.1 | 31.0 | 31.2 | **31.6** |
> | DQN | 8.1 | 14.3 | 22.4 | 24.3 | 25.6 | 26.4 | 27.9 | 29.0 | 30.0 | **31.6** |
> | Gittins | 19.1 | **25.8** | **28.6** | **30.3** | **30.8** | **31.1** | **31.3** | **31.6** | **31.6** | **31.6** |
>
> **Expected *undiscounted* reward for $\mathcal{Q}_{\theta, 1}$ on HIV network**
>
> | Percentage of nodes tested | 10% | 20% | 30% | 40% | 50% | 60% | 70% | 80% | 90% | 100% |
> | :-: | :-: | :-: | :-: | :-: | :-: | :-: | :-: | :-: | :-: | :-: |
> | Random | 5.8 | 11.4 | 13.7 | 19.8 | 22.5 | 24.7 | 26.4 | 28.1 | 29.7 | **31.6** |
> | Greedy | **16.6** | **22.3** | **23.8** | **24.8** | **26.7** | 27.6 | **30.1** | **30.8** | 31.2 | **31.6** |
> | DQN | 3.4 | 9.9 | 11.6 | 14.7 | 17.1 | 21.3 | 23.8 | 27.1 | 28.9 | **31.6** |
> | Gittins | 16.5 | 22.2 | **23.8** | **24.8** | 26.1 | **28.6** | 29.7 | 30.6 | **31.5** | **31.6** |
>
> From this preliminary set of experiments, we see that our proposed method (Gittins) remains competitive while the performance of all policies degrade as $\mathcal{Q}_{\theta,\varepsilon}$ becomes less representative of $\mathcal{P}$ due to larger $\varepsilon$ values. In our revision, we plan to conduct a broader suite of such experiments across all the disease networks and report them in the appendix.
>
> # Scalability and runtime
>
> We appreciate the concern regarding scalability. While the worst-case runtime of our Gittins index computation is $O(n^2 |\Omega|^2)$, this is tractable in our target applications. In our datasets, $|\Omega| = 2$ (binary test outcome), and graph sizes are typically $n \leq 10,000$ in public health studies (see [1,2,3]). As shown in Figure 5 and Appendix D.4, our method runs efficiently in practice and is the fastest among all considered baselines on larger graphs. Additionally, as noted on Lines 134–140 of our paper, we believe this is the first work to present an efficient method for computing Gittins indices in discrete branching bandits with history-dependent rewards.
>
> [1] Peter Bearman, James Moody, Katherine Stovel. *Chains of affection: The structure of adolescent romantic and sexual networks*. American Journal of Sociology, 2004.
>
> [2] Joel Wertheim, Sergei Kosakovsky Pond, Lisa Forgione, Sanjay Mehta, Ben Murrell, Sharmila Shah, Davey Smith, Konrad Scheffler, Lucia Torian. *Social and Genetic Networks of HIV-1 Transmission in New York City*. PLoS Pathogens, 2017.
>
> [3] A M Young, A B Jonas, U L Mullins, D S Halgin, J R Havens. *Network structure and the risk for HIV transmission among rural drug users*. AIDS and Behavior, 2013.
>
> # Clarification of $\phi_{X, b}$ and $\Phi_{Ch(X),b}$
>
> Thank you for raising this point. We agree that the distinction between $\phi_{X, b}$ and $\Phi_{\text{Ch}(X), b}$ could be clearer. To clarify:
>
> - $\phi_{X, b}$ represents the expected value of the subtree rooted at node $X$, given that its parent has label $b$.
> - $\Phi_{\text{Ch}(X), b}$ captures the expected value of the subtree *excluding $X$ itself*. That is, the contributions from the children of $X$, conditioned on $X$ having label $b$.
>
> We will make this distinction more explicit in our revision and improve the explanation in Lines 176–178.
>
> # Choice of deep RL baseline
>
> We appreciate the question regarding our choice of a single-step DQN as the deep RL baseline. While more recent graph-based RL and planning methods (e.g., policy gradient, Monte Carlo Tree Search) are indeed promising, they are generally data- and compute-intensive.
>
> In our problem application domain of disease testing, the available data is often limited to a few hundred/thousand individuals with one label each. This makes it challenging to effectively train more complex models. Additionally, such methods often require significant computational resources (e.g., GPU clusters), which are typically unavailable to public health agencies.
>
> Given these constraints, we focused on lightweight baselines like greedy and DQN that are both implementable and deployable in practice. As shown in Figure 5 and Appendix D.4, our method achieves strong performance while remaining efficient and practical.

---

> > ### Author Response · Authors · 2025-08-04
> >
> > Dear reviewer, I hope our response has clarified your doubts and concerns. Please let us know if there is anything else you would wish to discuss regarding our work.

---

### Official Review · Reviewer_DJzd · 2025-07-03

**Clarity:** 4
**Significance:** 3
**Originality:** 2
**Rating:** 3
**Confidence:** 4

**Summary:**

This work considers the problem of sequential decision-making over a graph where each node is associated with a label from a finite set of labels and a label-dependent
reward whose distribution is given by a Markov model. The goal is to find the optimal policy that collects the maximum expected discounted reward. The work investigates an action constraint defined here as the adaptive frontier exploration on graphs (AFEG) constraint where the next node to visit in the graph has to be chosen from the set of non-visited neighbours of the already visited nodes. The authors model network based disease testing using this framework motivating such a model in scenarios where tested individuals might be interviewed to trace contact with other individuals so that these individuals can subsequently be tested. The problem of network based disease testing is an important problem in public health and studies have shown better algorithms lead to better outcomes. In light of this, the current work reformulates AFEG as a branching bandit problem and proves optimality of Gittins index when the disease network is a forest. In case of general graphs, the authors use the BFS tree of the graph to run the Gittins index based policy, thus providing a heuristic for the general case. Moreover, the authors show that their algorithm runs in time polynomial in the size of the graph and the number of elements in the label space. Finally, the work demonstrates empirically the performance of Gittins index based policy for synthetic as well as real-world disease network datasets. The experiments show that Gittins index based policy is generally faster than other baselines while giving superior or competitive performance.

**Questions:**

See weaknesses above. Minor suggestion: please consider changing the notation $X_{\pi(S_t)}$ that indicates both label and the node currently. It would help the reader if this notation was disambiguated.

**Ethical Concerns:**

["NO or VERY MINOR ethics concerns only"]

**Final Justification:**

One of the claimed primary contribution of this work is the theoretical results, which does not appear novel and has been observed in the existing literature. Moreover, one of the theorems (Thm 6) seems to mis-specify the required conditions for the theorem to hold. However, to the authors' credit, the experimental results on real-world datasets seem promising. Therefore, I am inclined towards a borderline reject since although the theoretical parts are not novel, the experimental parts are promising and merits further research.

**Limitations:**

Yes

**Quality:**

2

**Strengths And Weaknesses:**

Strengths:

1. The formulation of network based disease testing as an AFEG problem is interesting. Network based disease testing is an important problem in public health and has shown merits in multiple studies. Therefore, this work is an important contribution in the literature of network based disease testing.

2. The computational efficiency results for Gittins index based policy which is generally intractable due to the curse of dimensionality is another novel aspect of the work.

3. The experiments performed are extensive with several real-world disease datasets considered on top of synthetic experiments. In most of the experiments on real data, the Gittins index based policy is either slightly superior or competitive. The experiment set up is also natural and well motivated.

4. Lastly, the writing is very clear and cogent with good organization and discussions making it easy for the reviewer to understand the work. Overall it was a pleasure to read.

Weaknesses:

1. Firstly, Theorem 1 is not very novel. It is a straightforward adaptation of the work [1]. Moreover, the guarantee works for the forest case which though is a good starting point, is theoretically a limiting assumption and may not hold in real world networks. However, to the authors’ defence the Gittins index based policy on the BFS tree of general graphs seems to give good empirical performance. Here however the question is why BFS tree and not other trees? This work might benefit if the authors can give provable guarantees on the global graph when Gittins based policy is run on an appropriately chosen tree of the graph.

2. The work considers a case where the label space is finite. The main computational result (Theorem 6) also banks on this assumption. This may be limiting in scenarios where richer data is available, for example in disease networks, not just discrete labels but also real valued intensity may be available. While general label spaces may be hard to compute, it would be interesting to see some generalization, for example, can it be done for continuous label spaces with an $\varepsilon$-discretization where the computational efficiency scale as $poly \log (\frac{1}{\varepsilon}$ whereas the optimal policy over the discretization may differ by $\varepsilon$ from optimal?

3. The current work relies on access to the graph structure and the distribution $\mathcal{P}$. While the assumption on graph structure is still fine, the assumption of access to distribution $\mathcal{P}$ exactly is a limiting one. Particularly, one of the baselines considered is DQN which relies only on sample access to $\mathcal{P}$. Instead of assuming exact oracle for $\mathcal{P}$, it would be interesting to see the theory developed for sample access to the distribution. Indeed, a more reinforcement learning style with finite sample convergence guarantees will be useful to the work. This is because currently the work uses existing data to first learn an approximate $\mathcal{P}$ and then use it for policy. However, this learning step makes certain strong structural assumptions which may be bypassed if an RL style algorithm can be given.

4. Figure 14 seems to show that the DQN policy is much better than the Gittins policy with the later having constant suboptimal gap for all the disease networks considered. This is in contradiction to the experimental results in the remaining paper. Moreover, the motivation for the smaller dataset created by subsampling with threshold $\tau = 300$ is not clear. Indeed, Fig. 14 result on the full dataset seems contradictory to the result on this subsample. Moreover, the authors’ algorithm does not seem to provide much performance benefit over greedy/DQN in the low $\beta$ regime on synthetic dataset. In light of this, it is not clear why the authors do not repeat experiments with low $\beta$ on the real world dataset.

[1] Godfrey Keller, Alison Oldale. Branching bandits: a sequential search process with correlated pay-offs, Journal of Economic Theory, Volume 113, Issue 2, 2003, Pages 302-315.

---

> ### Author Rebuttal · Authors · 2025-07-31
>
> Thank you for your time and thoughtful feedback on our submission. We sincerely appreciate your comments and the opportunity to address your concerns. We will incorporate a version of this discussion into our revised manuscript.
>
> # Theorem 3 and novelty
>
> We acknowledge that Theorem 3 is a direct consequence of Theorem 1 from [KO03]. We clarified this in the main text (Line 184). Our primary contribution in Section 3.1 is to establish that the connection that Gittins indices for branching bandits provide an optimal solution to AFEG instances when the input graph is a forest. Additionally, while [KO03] proves optimality, it does not provide an efficient computation method. Our contributions go further in two ways:
>
> 1. Proving new structural properties of the Gittins computation (Lemma 4 and Proposition 5)
> 2. Developing a polynomial-time algorithm with an accompanying implementation to compute Gittins indices (Theorem 6)
>
> # BFS tree projection
>
> We selected the BFS tree projection to minimize the depth of the induced tree, preserving proximity between nodes and root. This ensures that options at the frontier are not artificially constrained by long detours due to us removing edges to invoke Gittins.
>
> For example, consider the graph made of a small cycle $v_0 - v_1 - v_2 - v_0$ and a large cycle $v_1 - v_2 - v_3 - \ldots - v_k - v_1$, where $k \gg 3$. Suppose $v_0$ is the root. The induced BFS tree will keep the path $v_0 - v_2$ while another induced tree that removes edges $v_0 - v_2$ and $v_1 - v_2$, resulting in $v_2$ being reachable from the root only via a very long path $v_0 - v_1 - v_k - \ldots - v_3 - v_2$. It would be interesting to investigate different tree projections for other approaches when there is no frontier exploration constraint.
>
> # Extending to continuous label space
>
> We believe discrete labels are already meaningful in many applications, particularly in health domains. For example, positive/negative test results, or coarser categorical risk levels (e.g., low/medium/high), are common in practice.
>
> That said, we agree this is an important and interesting extension. However, even under $\varepsilon$-discretization, the analysis is nontrivial. This is because the function $\phi$ can have non-continuous derivatives, and $\Phi$ aggregates these through products and integrals. This makes bounding policy errors with respect to discretization challenging.
>
> # Oracle access to underlying probability distribution $\\mathcal{P}$
>
> We agree that the assumption of oracle access to $\mathcal{P}$ may not always hold in practice. In real-world settings, such as modeling the spread of sexually transmitted infections (STIs), it is reasonable to approximate the distribution $\mathcal{P}$ using a pairwise Markov Random Field (MRF) over the contact network $\mathcal{G}$. As you kindly noted, we outline in Appendix B how the parameters $\theta$ of such a model can be estimated from historical data via maximum pseudo-likelihood estimation (MPLE). In addition, Appendix E.3 contains an analysis of how errors in estimating $\mathcal{P}$ affect decision quality.
>
> In the original submission, we assumed that $\mathcal{P}\_{\theta}$ (estimated via MPLE) was the ground truth. In response to your comment, we have conducted additional experiments during the rebuttal period to assess robustness when the policy only has access to a noisy version $\mathcal{Q}\_{\theta}$ of the true distribution $\mathcal{P}_{\theta}$.
>
> We perturbed the learned parameters $\theta$ with noise of magnitude $\varepsilon \in \\{0.1, 0.3, 1\\}$ to simulate distributional mismatch, while keeping the evaluation grounded in the original $\mathcal{P}\_{\theta}$. For reference, in the HIV setting, we have $\ell_{\infty}(\theta_{\text{unary}}) = 1.43$ and $\ell_{\infty}(\theta_{\text{pairwise}}) = 1.75$. Thus, $\varepsilon = 1$ constitutes a significant perturbation.
>
> Since plots and external URLs are disallowed, we summarize our results in tabular form, reporting the expected undiscounted reward over a random subset of approximately 300 nodes in the HIV network (following the format in Appendix D.3). We focus on *undiscounted* reward as it better reflects the practical goals of maximizing the number of infected individuals detected under a fixed testing budget. For clarity, we **bold** the best-performing methods for each budget level.
>
> **Expected *undiscounted* reward for $\mathcal{Q}_{\theta, 0.1}$ on HIV network**
>
> | Percentage of nodes tested | 10% | 20% | 30% | 40% | 50% | 60% | 70% | 80% | 90% | 100% |
> | :-: | :-: | :-: | :-: | :-: | :-: | :-: | :-: | :-: | :-: | :-: |
> | Random | 6.1 | 11.4 | 16.0 | 19.4 | 22.3 | 24.6 | 26.6 | 28.2 | 29.8 | **31.6** |
> | Greedy | **20.1** | 25.8 | 26.6 | 26.8 | 27.0 | 27.5 | 30.1 | 31.0 | 31.2 | **31.6** |
> | DQN | 15.7 | 22.7 | 25.4 | 26.6 | 27.0 | 27.3 | 27.9 | 28.2 | 29.9 | **31.6** |
> | Gittins | **20.1** | **28.4** | **29.4** | **30.5** | **30.9** | **31.2** | **31.3** | **31.6** | **31.6** | **31.6** |
>
> **Expected *undiscounted* reward for $\mathcal{Q}_{\theta, 0.3}$ on HIV network**
>
> | Percentage of nodes tested | 10% | 20% | 30% | 40% | 50% | 60% | 70% | 80% | 90% | 100% |
> | :-: | :-: | :-: | :-: | :-: | :-: | :-: | :-: | :-: | :-: | :-: |
> | Random | 6.1 | 11.5 | 16.0 | 19.5 | 22.3 | 24.8 | 26.6 | 28.3 | 29.8 | **31.6** |
> | Greedy | **19.2** | 25.7 | 26.4 | 26.8 | 27.0 | 27.6 | 30.1 | 31.0 | 31.2 | **31.6** |
> | DQN | 8.1 | 14.3 | 22.4 | 24.3 | 25.6 | 26.4 | 27.9 | 29.0 | 30.0 | **31.6** |
> | Gittins | 19.1 | **25.8** | **28.6** | **30.3** | **30.8** | **31.1** | **31.3** | **31.6** | **31.6** | **31.6** |
>
> **Expected *undiscounted* reward for $\mathcal{Q}_{\theta, 1}$ on HIV network**
>
> | Percentage of nodes tested | 10% | 20% | 30% | 40% | 50% | 60% | 70% | 80% | 90% | 100% |
> | :-: | :-: | :-: | :-: | :-: | :-: | :-: | :-: | :-: | :-: | :-: |
> | Random | 5.8 | 11.4 | 13.7 | 19.8 | 22.5 | 24.7 | 26.4 | 28.1 | 29.7 | **31.6** |
> | Greedy | **16.6** | **22.3** | **23.8** | **24.8** | **26.7** | 27.6 | **30.1** | **30.8** | 31.2 | **31.6** |
> | DQN | 3.4 | 9.9 | 11.6 | 14.7 | 17.1 | 21.3 | 23.8 | 27.1 | 28.9 | **31.6** |
> | Gittins | 16.5 | 22.2 | **23.8** | **24.8** | 26.1 | **28.6** | 29.7 | 30.6 | **31.5** | **31.6** |
>
> From this preliminary set of experiments, we see that our proposed method (Gittins) remains competitive while the performance of all policies degrade as $\mathcal{Q}_{\theta,\varepsilon}$ becomes less representative of $\mathcal{P}$ due to larger $\varepsilon$ values. In our revision, we plan to conduct a broader suite of such experiments across all the disease networks and report them in the appendix.
>
> # Clarification regarding figure 14
>
> Thank you for pointing this out. The figure in question does **not** show DQN outperforming Gittins. Due to a rendering error, the legend was mislabeled. In this instance, Gittins corresponds to the yellow curve, and DQN results were omitted because each training run exceeded a full day. We will correct this in the final version.
>
> # Low $\beta$ on real-world dataset
>
> As noted in Lines 286–289, our goal for real-world scenarios is to identify as many positive cases as possible under a fixed testing budget. The discount factor $\beta$ is simply a tool to emphasize early discoveries; any $\beta \in (0,1)$ is valid. However, smaller $\beta$ values heavily penalize late discoveries, which can be counterproductive when detection at any point is valuable.
>
> To illustrate this, we re-ran HIV experiments under $\beta = 0.1$ and $\beta = 0.99$ and report the undiscounted performance in both cases. For easier comparison, we **bold** the largest reward attained at each level of budget available.
>
> **Expected *undiscounted* reward for $\beta = 0.1$ on HIV network**
>
> | Percentage of nodes tested | 10% | 20% | 30% | 40% | 50% | 60% | 70% | 80% | 90% | 100% |
> | :-: | :-: | :-: | :-: | :-: | :-: | :-: | :-: | :-: | :-: | :-: |
> | Random | 6.1 | 11.0 | 15.2 | 18.5 | 21.3 | 23.5 | 25.6 | 28.0 | 30.0 | **31.6** |
> | Greedy | 17.7 | **22.4** | **23.0** | 23.3 | 23.4 | 29.2 | 30.1 | **31.1** | **31.2** | **31.6** |
> | DQN | 16.9 | 22.2 | 22.9 | 23.2 | 23.7 | 24.2 | 24.8 | 30.6 | 31.2 | **31.6** |
> | Gittins | **17.8** | **22.4** | **23.0** | **29.2** | **29.8** | **30.6** | **30.9** | 31.0 | 31.0 | **31.6** |
>
> **Expected *undiscounted* reward for $\beta = 0.99$ on HIV network**
>
> | Percentage of nodes tested | 10% | 20% | 30% | 40% | 50% | 60% | 70% | 80% | 90% | 100% |
> | :-: | :-: | :-: | :-: | :-: | :-: | :-: | :-: | :-: | :-: | :-: |
> | Random | 6.0 | 11.6 | 15.7 | 19.1 | 21.7 | 23.9 | 25.9 | 28.1 | 30.0 | **31.6** |
> | Greedy | 17.7 | 22.4 | 23.0 | 23.3 | 23.4 | 29.2 | 30.1 | 31.1 | 31.2 | **31.6** |
> | DQN | 13.9 | 18.3 | 21.3 | 26.2 | 27.7 | 28.5 | 29.8 | 30.2 | 30.8 | **31.6** |
> | Gittins | **19.0** | **26.9** | **29.5** | **30.6** | **30.9** | **31.2** | **31.3** | **31.6** | **31.6** | **31.6** |
>
> Gittins maintains strong performance across the board and achieves higher coverage than baselines even when $\beta$ is small. However, note that a small $\beta$ effectively "zeroes out the reward" for an identification many steps in the future, resulting in degradation of policies that optimize long-term performance (such as DQN and Gittins).
>
> # Notation Overload
>
> We appreciate the reviewer's suggestion regarding potential ambiguity in our notation. We follow standard conventions in the graphical models literature, where variables and nodes are often used interchangeably, and the graph structure implicitly defines the factorization of the joint distribution. In this setting, it is common to use a symbol like $X$ to refer to both the node and its associated random variable, so that expressions like $X = v$ denote the event that variable $X$ takes value $v$. That said, we understand that readers from other communities may find this notation unfamiliar, and so we will include a brief remark early in the paper clarifying this convention to aid accessibility.

---

> > ### Author Response · Authors · 2025-08-04
> >
> > Dear reviewer, I hope our response has clarified your doubts and concerns. Please let us know if there is anything else you would wish to discuss regarding our work.

---

> ### Comment · Reviewer_DJzd · 2025-08-06
> **Response to rebuttal**
>
> I would like to thank the authors for their clarifications. I believe the contribution of this paper to be primarily experimental, specifically, formulating network-based disease testing as an AFEG (branching bandit) problem and applying Gittins-index based algorithm to this problem for real world datasets. The theoretical aspects of this work are not that novel, therefore, the contribution 1 of the paper bears less weightage to me. I feel that focusing and strengthening contribution 2 and 3 will be more beneficial to the current work. The detailed concerns are as follows.
>
> Theorem 3 and novelty: Lemma 4 is already observed as a remark by KO03 in their proof of theorem 1 (see the paragraph between eqn 2 and 3). Proposition 5 does not seem related to the current paper, and indeed the authors mention it as some additional properties they prove. Therefore, the theoretical contribution of this work is marginal. The only important theoretical contribution I feel this paper makes is Theorem 6. However, once the authors note Lemma 4 (observation in KO03), this follows directly via a dynamic programming argument. Moreover, in the proof of Theorem 6, the authors assume “any conditional probability value can be obtained in constant time via oracle access to P” (line 802-803) whereas in the theorem statement, the assumption is oracle access to joint distribution P. Can the authors lay out how to construct a conditional probability model in constant time out of a joint distribution? Otherwise, the theorem statement is misleading.
>
> Continuous Label Space: Note that $\phi$ is a piecewise linear function with finitely many pieces, so even if its derivative is non-continuous, it is piecewise constant. With the other theoretical results in the paper seeming limited to me, I would like to encourage the authors to investigate this direction as an interesting theory question.
>
> Experimental Results: The experimental results are promising even under perturbed $\theta$. However, note that this does not fully address my concern, which was regarding an (Gittins-index based) algorithm that works with sample access to the distribution. The current set of experiments presented by the authors in the rebuttal only backs up their theory in Appendix E.3 (Lemma 8), which was never my concern. To reiterate, this work might benefit from an algorithm that can work with samples from the distribution to construct (noisy) Gittins index.

---

> > ### Author Response · Authors · 2025-08-06
> > **Part 1/2**
> >
> > Thank you very much for your thoughtful follow-up and for engaging further with our work. While we value your feedback as it helps sharpen both our understanding and presentation, we would like to respectfully push back on the characterization that our contributions are primarily experimental, and that the theoretical components are of limited novelty.
> >
> > Our work aims to bridge a practically motivated and messy real-world problem (network-based disease testing) with a principled theoretical model via a novel reduction to branching bandits. This style of contribution, casting complex domains into well-founded mathematical frameworks, has been recognized as foundational across multiple areas of AI and operations research (e.g., casting security operations as Stackelberg games [1,2,3,4] or health resource management as restless bandits [5,6,7]). We see our work in the same spirit, and believe that our reduction, together with the structural insights and implementation advances it enables, is both non-trivial and impactful.
> >
> > # Theoretical Foundations and Novel Reduction
> >
> > We view our theoretical contributions as comprising the following:
> >
> > 1. *Efficient computation*: While [KO03] proved the optimality of Gittins indices for branching bandits, no efficient implementation has previously been proposed. As noted in Lines 134--137 of our paper:
> >
> > > While Gittins index policies are known to be optimal for branching bandits [KO03], no efficient method has been proposed to compute them in general. Indeed, Gittins indices are underused in practice due to perceived computational intractability in all but simple settings [Sco10, MKLL12, Edw19].
> >
> > Our work fills this gap by developing a polynomial-time and polynomial-space implementation (Theorem 6), supported by a publicly available codebase and demonstrated on real-world graphs. The absence of an efficient computational method for this Gittins index over the last 20 years reflects the nontrivial nature of this contribution.
> >
> > 2. *New structural insights*: We respectfully disagree with the assertion that Lemma 4 is already fully observed in [KO03]. While that work notes convexity and monotonicity between Equations 2 and 3, it does not state or prove that the value functions are piecewise linear, a property we prove explicitly and rely on for efficient recursion.
> >
> > Similarly, we would like to clarify a misunderstanding regarding Proposition 5. The reviewer noted that it "does not seem related to the current paper", but it is in fact an essential component of our implementation. The first bullet of Proposition 5 is directly exploited in our recursive Gittins computation (see `_build_Phi` in `gittins_policy.py`), and enables a key simplification in computing composed value functions. When we state that these results may be of independent interest, we do not mean they are unrelated to our work. Rather, we mean that the structural properties we prove apply more broadly than just our setting, while still being crucial for our algorithmic design and analysis. We hope this clears up the confusion.
> >
> > 3. *Structural reduction*: We introduce AFEG, a formalization of the network-based testing problem, and prove that tree-structured AFEG instances can be solved optimally via the Gittins index policy for branching bandits. To our knowledge, this connection is new and enables the application of classical bandit theory in a fresh, practically relevant domain. In the same spirit as how Stackelberg's model of competing firms from the 1930s later became foundational to the real-world deployment of Stackelberg Security Games (SSGs) in domains like airport and wildlife protection, we believe that drawing such principled connections between classical models and modern applications is a meaningful contribution in its own right.
> >
> > # Clarification on Theorem 6
> >
> > Thank you for pointing this out. Theorem 6 aimed to abstract away the cost of computing conditional probabilities of the form $\mathcal{P}(\mathbf{x} \mid \mathbf{y})$, and focuses on bounding the number of recursive computations required for Gittins index evaluation. In our tree setting, the conditioning set contains at most one variable, so the required conditional probabilities can be derived from the joint distribution $\mathcal{P}$ via marginalization with minimal overhead. We will clarify this in our revision to better align the theorem statement with the assumptions in the proof.

---

> ### Author Response · Authors · 2025-08-06
> **Part 2/2**
>
> # Continuous label space
>
> We agree that extending the framework to continuous label spaces is an interesting direction for future work. Our current focus addresses discrete label settings, and already generalizes beyond the binary case of our real-world motivation of disease testing. Further extensions to continuous domains would be valuable, but are beyond the natural scope of this work and not necessary to demonstrate the strength of our current contributions.
>
> # Sample-based access vs. model-based estimation
>
> Thank you for clarifying this point. Developing Gittins-like policies that operate directly from sample-based access is indeed an interesting direction. Our current approach uses a learned model to compute exact indices and achieves strong empirical performance. Exploring sample-based variants is complementary to our framework, but beyond the intended scope of this paper.
>
> # Final remarks
>
> We appreciate your interest in directions such as continuous label spaces and sample-based estimation. While we agree that these are intellectually interesting, we respectfully view these as natural next steps, rather than missing components of the current paper. The core contribution of this work --- its problem formulation, structural reduction, efficient algorithm design, and strong empirical performance --- is, in our view, already substantial and meets the bar for publication on its own merit. Requiring these additional components would significantly broaden the scope beyond what is typical for a single conference paper.
>
> # References
>
> [1] Avrim Blum, Nika Haghtalab, Ariel D. Procaccia. "Learning optimal commitment to overcome insecurity". NeurIPS, 2014.
>
> [2] Blog post by Ariel D. Procaccia: https://agtb.wordpress.com/2012/06/17/is-game-theory-useful/
>
> [3] Maria-Florina Balcan, Avrim Blum, Nika Haghtalab, Ariel D. Procaccia. "Commitment Without Regrets: Online Learning in Stackelberg Security Games". EC, 2015.
>
> [4] A more recent work: Maria-Florina Balcan, Kiriaki Fragkia, and Keegan Harris. "Learning in Structured Stackelberg Games". arXiv, 2025.
>
> [5] Aditya Mate, Jackson A. Killian, Haifeng Xu, Andrew Perrault, Milind Tambe. "Collapsing bandits and their application to public health intervention". NeurIPS, 2020.
>
> [6] Dexun Li, Pradeep Varakantham. "Efficient resource allocation with fairness constraints in restless multi-armed bandits". UAI, 2022.
>
> [7] Archit Sood, Shweta Jain, Sujit Gujar. "Fairness of Exposure in Online Restless Multi-armed Bandits". AAMAS, 2024.

---

> > ### Comment · Reviewer_DJzd · 2025-08-07
> > **Response to authors**
> >
> > Thanks for your reply and the further explanation.
> >
> > Regarding new structural insights, and the claim that “While that work notes convexity and monotonicity between Equations 2 and 3, it does not state or prove that the value functions are piecewise linear, a property we prove explicitly and rely on for efficient recursion.” - KO03 already notes: “It is fairly easy to show that $\Phi(M,X)$ as a function of $M$ is non-decreasing and convex (convexity following from the fact that we are dealing with the supremum of expressions which are linear in $M$)” From this, it is clear that $\Phi(M,X)$ is piecewise linear in $M$ since it is a well-known fact that supremum of piecewise linear functions is convex. Therefore, I am not convinced about the novelty of the structural result. Further, given this piecewise linearity, efficient computation follows from dynamic programming arguments since the paper is dealing with trees where one can start filling the DP table from the leaves (which cannot be done in general graphs because of absence of a leaf node). This further leads me to believe that the computational benefit comes from the tree structure, and not from the piecewise linearity observation which is present in the definition of Gittins index itself and noted in KO03.
> >
> > Regarding Prop 5, if it is being used in implementation, I suggest it should also be made part of the pseudocode?
> >
> > Regarding point 3, I agree that the connection is new and merits further research.
> >
> > Regarding theorem 6: “In our tree setting, the conditioning set contains at most one variable, so the required conditional probabilities can be derived from the joint distribution $\mathcal{P}$ via marginalization with minimal overhead.” I do not immediately see how the bold part can be computed “with minimal overhead”. For example, if $\mathcal{P}$ is a joint distribution of $4$ variables $W,X,Y,Z$, and the support size of each variable is $m$, then to obtain the marginal $\mathcal{P}(W,X)$ from the joint distribution $\mathcal{P}(W,X,Y,Z)$, one needs $m^2$ computations (so as to compute $\mathcal{P}(W=w, X=x) = \sum_{y,z} \mathcal{P}(W=w, X=x, Y=y, Z=z)$ (observe that the summation runs over $m^2$ things). The tree setting does not help since you still have to obtain the marginal of the leaf variable which will require $m^(n-1)$ computations to marginalize out all but the leaf variable, which is exponential time!

---

### Official Review · Reviewer_PxU7 · 2025-07-05

**Clarity:** 4
**Significance:** 4
**Originality:** 3
**Rating:** 5
**Confidence:** 4

**Summary:**

Consider the problem of sequential decision making on a graph, where the arms that can be pulled are only be neighbours of previously pulled arms. This is the frontier of exploration. Specifically, this paper addresses the "Adaptive Frontier Exploration on Graphs" (AFEG) problem, a sequential decision-making task relevant to high-impact applications like network-based disease testing.

The core contribution is a novel, efficient, polynomial-time algorithm for computing the Gittins index policy, which is provably optimal for AFEG instances on tree-structured graphs. The authors connect this deep theoretical result to a pressing real-world problem, demonstrating on real-world HIV interaction networks that their method can improve testing efficiency compared to strong baselines, including a DQN (deep Q net) approach. The work is technically deep, rigorously presented, and its claims are backed by strong empirical evidence.

**Questions:**

- Are real world graphs truly forests (for which the optimality results hold)? In the real world, are people not fully connected to each other?
- Where does this come from (line 701 in appendix) $θ_1 \in R^{2+2d}$ and $θ_2 \in R^{4+5d}$
- How valid is the assumption on non-decreasing piecewise linear function?

**Ethical Concerns:**

["NO or VERY MINOR ethics concerns only"]

**Final Justification:**

maintain score.

**Limitations:**

Please include a more detailed limitations section in your work. Since this paper has potential to be applied to medical domains, it would be prudent to state the limitations in greater detail to potential users.

**Quality:**

4

**Strengths And Weaknesses:**

### Strengths
1. One of the papers I was excited to read, and I've been reading MAB papers for a long time! Very interesting and practical problem formulation.
2. The paper's primary contribution is arguably proving the first poliynomial time algorithm for the provably optimal policy in discrete branching bandits. This is a foundational contribution to the MAB and RL literature, with implications beyond the disease testing/contact tracing problem.
3. I was very impressed with the empirical results which indicate that nearly all HIV-positive individuals were identified by testing only half the population in the ICPSR dataset, and for example works better than DQN (deep Q networks). Given decreasing medical and research budgets, this could be an important method to release to real-world use cases.

### Weakness

The model assumes an oracle access to joint probability distribution P, which can provide answers about the conditional probability distributions over Bayesian Nets.

Note: But in Appendix B in the paper, they convert the weakness into a strength by actually building this oracle out from a model using maximum pseudo-likelihood estimation (MPLE). This makes the oracle assumption plausible. Yet the authors acknowledge that such a parameter fitting may not recover the exact real world dynamics (Line 631).

---

> ### Author Rebuttal · Authors · 2025-07-31
>
> Thank you for your time and thoughtful feedback on our submission. We sincerely appreciate your comments and the opportunity to address your concerns. Below, we respond to the main points raised in the review and will incorporate a version of this discussion into our revised manuscript.
>
> # Oracle access to underlying probability distribution $\\mathcal{P}$
>
> We agree that the assumption of oracle access to $\mathcal{P}$ may not always hold in practice. In real-world settings, such as modeling the spread of sexually transmitted infections (STIs), it is reasonable to approximate the distribution $\mathcal{P}$ using a pairwise Markov Random Field (MRF) over the contact network $\mathcal{G}$. As you kindly noted, we outline in Appendix B how the parameters $\theta$ of such a model can be estimated from historical data via maximum pseudo-likelihood estimation (MPLE). In addition, Appendix E.3 contains an analysis of how errors in estimating $\mathcal{P}$ affect decision quality.
>
> In the original submission, we assumed that $\mathcal{P}\_{\theta}$ (estimated via MPLE) was the ground truth. In response to your comment, we have conducted additional experiments during the rebuttal period to assess robustness when the policy only has access to a noisy version $\mathcal{Q}\_{\theta}$ of the true distribution $\mathcal{P}_{\theta}$.
>
> We perturbed the learned parameters $\theta$ with noise of magnitude $\varepsilon \in \\{0.1, 0.3, 1\\}$ to simulate distributional mismatch, while keeping the evaluation grounded in the original $\mathcal{P}\_{\theta}$. For reference, in the HIV setting, we have $\ell_{\infty}(\theta_{\text{unary}}) = 1.43$ and $\ell_{\infty}(\theta_{\text{pairwise}}) = 1.75$. Thus, $\varepsilon = 1$ constitutes a significant perturbation.
>
> Since plots and external URLs are disallowed, we summarize our results in tabular form, reporting the expected undiscounted reward over a random subset of approximately 300 nodes in the HIV network (following the format in Appendix D.3). We focus on *undiscounted* reward as it better reflects the practical goals of maximizing the number of infected individuals detected under a fixed testing budget. For clarity, we **bold** the best-performing methods for each budget level.
>
> **Expected *undiscounted* reward for $\mathcal{Q}_{\theta, 0.1}$ on HIV network**
>
> | Percentage of nodes tested | 10% | 20% | 30% | 40% | 50% | 60% | 70% | 80% | 90% | 100% |
> | :-: | :-: | :-: | :-: | :-: | :-: | :-: | :-: | :-: | :-: | :-: |
> | Random | 6.1 | 11.4 | 16.0 | 19.4 | 22.3 | 24.6 | 26.6 | 28.2 | 29.8 | **31.6** |
> | Greedy | **20.1** | 25.8 | 26.6 | 26.8 | 27.0 | 27.5 | 30.1 | 31.0 | 31.2 | **31.6** |
> | DQN | 15.7 | 22.7 | 25.4 | 26.6 | 27.0 | 27.3 | 27.9 | 28.2 | 29.9 | **31.6** |
> | Gittins | **20.1** | **28.4** | **29.4** | **30.5** | **30.9** | **31.2** | **31.3** | **31.6** | **31.6** | **31.6** |
>
> **Expected *undiscounted* reward for $\mathcal{Q}_{\theta, 0.3}$ on HIV network**
>
> | Percentage of nodes tested | 10% | 20% | 30% | 40% | 50% | 60% | 70% | 80% | 90% | 100% |
> | :-: | :-: | :-: | :-: | :-: | :-: | :-: | :-: | :-: | :-: | :-: |
> | Random | 6.1 | 11.5 | 16.0 | 19.5 | 22.3 | 24.8 | 26.6 | 28.3 | 29.8 | **31.6** |
> | Greedy | **19.2** | 25.7 | 26.4 | 26.8 | 27.0 | 27.6 | 30.1 | 31.0 | 31.2 | **31.6** |
> | DQN | 8.1 | 14.3 | 22.4 | 24.3 | 25.6 | 26.4 | 27.9 | 29.0 | 30.0 | **31.6** |
> | Gittins | 19.1 | **25.8** | **28.6** | **30.3** | **30.8** | **31.1** | **31.3** | **31.6** | **31.6** | **31.6** |
>
> **Expected *undiscounted* reward for $\mathcal{Q}_{\theta, 1}$ on HIV network**
>
> | Percentage of nodes tested | 10% | 20% | 30% | 40% | 50% | 60% | 70% | 80% | 90% | 100% |
> | :-: | :-: | :-: | :-: | :-: | :-: | :-: | :-: | :-: | :-: | :-: |
> | Random | 5.8 | 11.4 | 13.7 | 19.8 | 22.5 | 24.7 | 26.4 | 28.1 | 29.7 | **31.6** |
> | Greedy | **16.6** | **22.3** | **23.8** | **24.8** | **26.7** | 27.6 | **30.1** | **30.8** | 31.2 | **31.6** |
> | DQN | 3.4 | 9.9 | 11.6 | 14.7 | 17.1 | 21.3 | 23.8 | 27.1 | 28.9 | **31.6** |
> | Gittins | 16.5 | 22.2 | **23.8** | **24.8** | 26.1 | **28.6** | 29.7 | 30.6 | **31.5** | **31.6** |
>
> From this preliminary set of experiments, we see that our proposed method (Gittins) remains competitive while the performance of all policies degrade as $\mathcal{Q}_{\theta,\varepsilon}$ becomes less representative of $\mathcal{P}$ due to larger $\varepsilon$ values. In our revision, we plan to conduct a broader suite of such experiments across all the disease networks and report them in the appendix.
>
> # Structure of real-world graphs
>
> We agree that real-world graphs are not truly forests (see Line 210). However, empirical studies have shown that STI transmission graphs are often sparse and exhibit tree-like structure (e.g., [1,2,3]). While certain high-degree nodes (e.g., sex workers) may exist, the network as a whole is far from fully connected. We refer to Appendix D.2 for an empirical evaluation of how performance degrades as the graph departs from tree structure.
>
> [1] Peter Bearman, James Moody, Katherine Stovel. *Chains of affection: The structure of adolescent romantic and sexual networks*. American Journal of Sociology, 2004.
>
> [2] Joel Wertheim, Sergei Kosakovsky Pond, Lisa Forgione, Sanjay Mehta, Ben Murrell, Sharmila Shah, Davey Smith, Konrad Scheffler, Lucia Torian. *Social and Genetic Networks of HIV-1 Transmission in New York City*. PLoS Pathogens, 2017.
>
> [3] A M Young, A B Jonas, U L Mullins, D S Halgin, J R Havens. *Network structure and the risk for HIV transmission among rural drug users*. AIDS and Behavior, 2013.
>
> # $\theta_1$ and $\theta_2$
>
> $\theta_1$ and $\theta_2$ are the parameter vectors of the unary and pairwise potentials in the pairwise-MRF model described in Appendix B.1. These parameters correspond to feature maps $f_1$ and $f_2$ from Equations (4) and (5), which consist of monomials (up to second order) of the covariate features. The construction respects symmetry and the maximum entropy principle, ensuring proper dimension alignment between features and parameters.
>
> # Non-decreasing piecewise linear function
>
> The non-decreasing piecewise linearity is **not** an assumption but a provable property of the Gittins index computation, as shown in Lemma 4 (line 192) and Proposition 5 (line 201). The detailed proof appears in Appendix E.1.
>
> # Limitations
>
> As mentioned in Section 5, we have provided additional discussion on broader impact and limitations in Appendix C.

---

### Decision · Program_Chairs · 2025-09-17

**Decision:**

Accept (poster)

**Comment:**

Summary: This paper introduces the Adaptive Frontier Exploration on Graphs (AFEG) problem, motivated by network-based disease testing, and develops an efficient Gittins index-based policy for it. The authors prove the policy is optimal for tree-structured graphs and demonstrate its strong empirical performance against baselines on both synthetic and real-world public health data.

Strengths: The paper's main strengths, as noted by all reviewers, are its practical problem formulation, its extensive empirical results on a real-world application, and its delivery of the first practical, polynomial-time algorithm for computing Gittins indices in this important class of branching bandit problems.

Weaknesses: The main weakness, highlighted during a detailed discussion with one reviewer, is that the core mathematical property (piecewise linearity) enabling the algorithm is a straightforward derivation from prior work, raising valid questions about the depth of the theoretical novelty.

Reasons for Decision: The decision to accept this paper is based on the judgment that its significant algorithmic contribution outweighs the concerns about pure theoretical novelty. The authors successfully designed the first efficient and practical algorithm for a problem that has been theoretically understood but computationally unsolved for two decades. This non-trivial leap from theory to a usable method, supported by an application and empirical evidence, constitutes a valid contribution to the field.

Discussion Process: While a high-level debate on the definition of novelty with one reviewer remained unresolved, the discussion supports the view that the paper's main contribution is a non-trivial algorithmic advance. The authors also successfully addressed other concerns, such as the assumption of oracle access to the data distribution. The authors will be expected to revise Theorem 6 in the final version to more precisely state their assumptions about the underlying probabilistic model, as discussed during the review process.